# Improving Predictor Reliability with Selective Recalibration

**Thomas P. Zollo**                                            *tpz2105@columbia.edu*
*Columbia University*

**Zhun Deng**                                                  *zhun.d@columbia.edu*
*Columbia University*

**Jake C. Snell**                                              *js2523@princeton.edu*
*Princeton University*

**Toniann Pitassi**                                            *toni@cs.columbia.edu*
*Columbia University*

**Richard Zemel**                                              *zemel@cs.columbia.edu*
*Columbia University*

**Reviewed on OpenReview:** *https://openreview.net/forum?id=Aoj9H6jl6F*

## Abstract

A reliable deep learning system should be able to accurately express its confidence with respect to its predictions, a quality known as calibration. One of the most effective ways to produce reliable confidence estimates with a pre-trained model is by applying a post-hoc recalibration method. Popular recalibration methods like temperature scaling are typically fit on a small amount of data and work in the model's output space, as opposed to the more expressive feature embedding space, and thus usually have only one or a handful of parameters. However, the target distribution to which they are applied is often complex and difficult to fit well with such a function. To this end we propose *selective recalibration*, where a selection model learns to reject some user-chosen proportion of the data in order to allow the recalibrator to focus on regions of the input space that can be well-captured by such a model. We provide theoretical analysis to motivate our algorithm, and test our method through comprehensive experiments on difficult medical imaging and zero-shot classification tasks. Our results show that selective recalibration consistently leads to significantly lower calibration error than a wide range of selection and recalibration baselines.

## 1 Introduction

In order to build user trust in a machine learning system, it is important that the system can accurately express its confidence with respect to its own predictions. Under the notion of calibration common in deep learning (Guo et al., 2017; Minderer et al., 2021), a confidence estimate output by a model should be as close as possible to the expected accuracy of the prediction. For instance, a prediction assigned 30% confidence should be correct 30% of the time. This is especially important in risk-sensitive settings such as medical diagnosis, where binary predictions alone are not useful since a 30% chance of disease must be treated differently than a 1% chance. While advancements in neural network architecture and training have brought improvements in calibration as compared to previous methods (Minderer et al., 2021), neural networks still suffer from miscalibration, usually in the form of overconfidence (Guo et al., 2017; Wang et al., 2021). In addition, these models are often applied to complex data distributions, possibly including outliers, and may have different calibration error within and between different subsets in the data (Ovadia et al., 2019; Perez-Lebel et al., 2023). We illustrate this setting in Figure 1a with a Reliability Diagram, a tool for visualizing

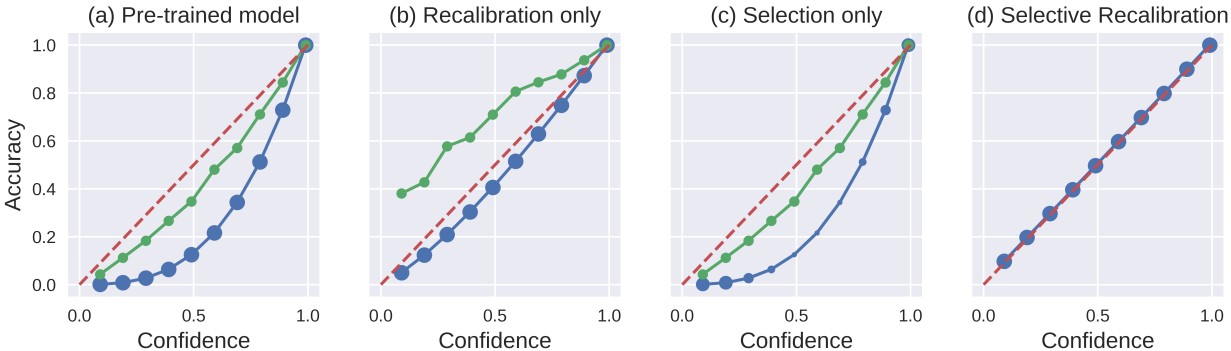

Figure 1: Reliability Diagrams for a model that has different calibration error (deviation from the diagonal) in different subsets of the data (here shown in blue and green). The data per subset is binned based on confidence values; each marker represents a bin, and its size depicts the amount of data in the bin. The red dashed diagonal represents perfect calibration, where confidence equals expected accuracy.

calibration by plotting average confidence against accuracy for bins of datapoints with similar confidence estimates.

To address this calibration error, the confidence estimates of a pre-trained model can be refined using a post-hoc recalibration method like Platt scaling (Platt, 1999), temperature scaling (Guo et al., 2017), or histogram binning (Zadrozny & Elkan, 2001). Given existing empirical evidence (Guo et al., 2017) and the fact that they are typically fit on small validation sets (on the order of hundreds to a few thousand examples), these "recalibrators" usually reduce the input space to the model's logits (e.g., temperature scaling) or predicted class scores (e.g., Platt scaling, histogram binning), as opposed to the high-dimensional and expressive feature embedding space. Accordingly, they are generally inexpressive models, having only one or a handful of parameters (Platt, 1999; Guo et al., 2017; Zadrozny & Elkan, 2001; Kumar et al., 2019). But the complex data distributions to which neural networks are often applied are difficult to fit well with such simple functions, and calibration error can even be exacerbated for some regions of the input space, especially when the model has only a single scaling parameter (see Figure 1b).

Motivated by these observations, we contend that these popular recalibration methods are a natural fit for use with a selection model. Selection models (El-Yaniv & Wiener, 2010; Geifman & El-Yaniv, 2017) are used alongside a classifier, and may reject a portion of the classifier's predictions in order to improve some performance metric on the subset of accepted (i.e., unrejected) examples. While selection models have typically been applied to improve classifier accuracy (Geifman & El-Yaniv, 2017), they have also been used to improve calibration error by rejecting the confidence estimates of a fixed model (Fisch et al., 2022). However, selection alone cannot fully address the underlying miscalibration because it does not alter the confidence output of the model (see Figure 1c), and the connection between selection and post-hoc recalibration remains largely unexplored.

In this work we propose *selective recalibration*, where a selection model and a recalibration model are jointly optimized in order to produce predictions with low calibration error. By rejecting some portion of the data, the system can focus on a region that can be well-captured by a simple recalibration model, leading to a set of predictions with a lower calibration error than under recalibration or selection alone (see Figure 1d). This approach is especially important when a machine learning model is deployed for decision-making in risk-sensitive domains such as healthcare, finance, and law, where a predictor must be well-calibrated if a human expert is to use its output to improve outcomes and avoid causing active harm. To summarize our contributions:

- We formulate selective recalibration, and offer a new loss function for training such a system, Selective Top-Label Binary Cross Entropy (S-TLBCE), which aligns with the typical notion of loss under smooth recalibrators like Platt or temperature scaling models.

- We test selective recalibration and S-TLBCE in real-world medical diagnosis and image classification experiments, and find that selective recalibration with S-TLBCE consistently leads to significantly lower calibration error than a wide range of selection and recalibration baselines.

- We provide theoretical insight to support our motivations and algorithm.

## 2  Related Work

Making well-calibrated predictions is a key feature of a reliable statistical model (Guo et al., 2017; Hebert-Johnson et al., 2018; Minderer et al., 2021; Fisch et al., 2022). Popular methods for improving calibration error given a pretrained model and labeled validation dataset include Platt (Platt, 1999) and temperature scaling (Guo et al., 2017), histogram binning (Zadrozny & Elkan, 2001), and Platt binning (Kumar et al., 2019) (as well as others like those in Naeini et al. (2015); Zhang et al. (2020)). Loss functions have also been proposed to improve the calibration error of a neural network in training, including Maximum Mean Calibration Error (MMCE) (Kumar et al., 2018), S-AvUC, and SB-ECE (Karandikar et al., 2021). Calibration error is typically measured using quantities such as Expected Calibration Error (Naeini et al., 2015; Guo et al., 2017), Maximum Calibration Error (Naeini et al., 2015; Guo et al., 2017), or Brier Score (Brier, 1950) that measure whether prediction confidence matches expected outcomes. Previous research (Roelofs et al., 2022) has demonstrated that calibration measures calculated using binning have lower bias when computed using equal-mass bins.

Another technique for improving ML system reliability is selective classification (Geifman & El-Yaniv, 2017; El-Yaniv & Wiener, 2010), wherein a model is given the option to abstain from making a prediction on certain examples (often based on confidence or out-of-distribution measures). Selective classification has been well-studied in the context of neural networks (Geifman & El-Yaniv, 2017; 2019; Madras et al., 2018). It has been shown to increase disparities in accuracy across groups (Jones et al., 2021), although work has been done to mitigate this effect in both classification (Jones et al., 2021) and regression (Shah et al., 2022) tasks by enforcing calibration across groups.

Recent work by Fisch et al. (2022) introduces the setting of calibrated selective classification, in which predictions from a pre-trained model are rejected for the sake of improving selective calibration error. The authors propose a method for training a selective calibration model using an S-MMCE loss function derived from the work of Kumar et al. (2018). Our work differs from this and other previous work by considering the *joint* training and application of selection and recalibration models. While Fisch et al. (2022) apply selection directly to a frozen model's outputs, we contend that the value in our algorithm lies in this joint optimization. Also, instead of using S-MMCE, we propose a new loss function, S-TLBCE, which more closely aligns with the objective function for Platt and temperature scaling.

Besides calibration and selection, there are other approaches to quantifying and addressing the uncertainty in modern neural networks. One popular approach is the use of ensembles, where multiple models are trained and their joint outputs are used to estimate uncertainty. Ensembles have been shown to both improve accuracy and provide a means to estimate predictive uncertainty without the need for Bayesian modeling (Lakshminarayanan et al., 2017). Bayesian neural networks (BNNs) offer an alternative by explicitly modeling uncertainty through distributions over the weights of the network, thus providing a principled uncertainty estimation (Blundell et al., 2015). Dropout can also be viewed as approximate Bayesian inference (Gal, 2016). Another technique which has received interest recently is conformal prediction, which uses past data to determine a prediction interval or set in which future points are predicted to fall with high probability (Shafer & Vovk, 2008; Vovk et al., 2005). Such distribution-free guarantees have been extended to cover a wide set of risk measures (Deng et al., 2023; Snell et al., 2023) and applications such as robot planning (Ren et al., 2023) and prompting a large language model (Zollo et al., 2024).

## 3  Background

Consider the multi-class classification setting with $K$ classes and data instances $(x, y) \sim \mathcal{D}$, where $x$ is the input and $y \in \{1, 2, ..., K\}$ is the ground truth class label. For a black box predictor $f$, $f(x) \in \mathbb{R}^K$ is a

vector where $f(x)_k$ is the predicted probability that input $x$ has label $k$; we denote the confidence in the top predicted label as $\hat{f}(x) = \max_k f(x)_k$. Further, we may access the unnormalized class scores $f_s(x) \in \mathbb{R}^K$ (which may be negative) and the feature embeddings $f_e(x) \in \mathbb{R}^d$. The predicted class is $\hat{y} = \arg\max_k f(x)_k$ and the correctness of a prediction is $y^c = \mathbf{1}\{y = \hat{y}\}$.

## 3.1 Selective Classification

In selective classification, a *selection model $g$* produces binary outputs, where 0 indicates rejection and 1 indicates acceptance. A common goal is to decrease some error metric by rejecting no more than a $1 - \beta$ proportion of the data for given target coverage level $\beta$. One popular choice for input to $g$ is the feature embedding $f_e(x)$, although other representations may be used. Often, a soft selection model $\hat{g} : \mathbb{R}^d \to [0, 1]$ is trained and $g$ is produced at inference time by choosing a threshold $\tau$ on $\hat{g}$ to achieve coverage level $\beta$ (i.e., $\mathbb{E}[\mathbf{1}\{\hat{g}(X) \geq \tau\}] = \beta$).

## 3.2 Calibration

The model $f$ is said to be top-label calibrated if $\mathbb{E}_{\mathcal{D}}[y^c|\hat{f}(x) = p] = p$ for all $p \in [0, 1]$ in the range of $\hat{f}(x)$. To measure deviation from this condition, we calculate expected calibration error (ECE):

$$\text{ECE}_q = \left( \mathbb{E}_{\hat{f}(x)} \left[ \left( \left| \mathbb{E}_{\mathcal{D}}[y^c|\hat{f}(x)] - \hat{f}(x) \right| \right)^q \right] \right)^{\frac{1}{q}}, \tag{1}$$

where $q$ is typically 1 or 2. A *recalibrator model $h$* can be applied to $f$ to produce outputs in the interval $[0, 1]$ such that $h(f(x)) \in \mathbb{R}^K$ is the recalibrated prediction confidence for input $x$ and $\hat{h}(f(x)) = \max_k h(f(x))_k$. See Section 4.3 for details on some specific forms of $h(\cdot)$.

## 3.3 Selective Calibration

Under the notion of calibrated selective classification introduced by Fisch et al. (2022), a predictor is selectively calibrated if $\mathbb{E}_{\mathcal{D}}\left[y^c|\hat{f}(x) = p, g(x) = 1\right] = p$ for all $p \in [0, 1]$ in the range of $\hat{f}(x)$ where $g(x) = 1$. To interpret this statement, for the subset of examples that are accepted (i.e., $g(x) = 1$), for a given confidence level $p$ the predicted label should be correct for $p$ proportion of instances. Selective expected calibration error is then calculated as:

$$\text{S-ECE}_q = \left( \mathbb{E}_{\hat{f}(x)} \left[ \left( \left| \mathbb{E}_{\mathcal{D}}[y^c|\hat{f}(x), g(x) = 1] - \hat{f}(x) \right| \right)^q \mid g(x) = 1 \right] \right)^{\frac{1}{q}}. \tag{2}$$

It should be noted that selective calibration is a separate goal from selective accuracy, and enforcing it may in some cases decrease accuracy. For example, a system may reject datapoints with $\hat{f}(x) = 0.7$ and $p(y^c = 1|x) = 0.99$ (which will be accurate 99% of the time) in order to retain datapoints with $\hat{f}(x) = 0.7$ and $p(y^c = 1|x) = 0.7$ (which will be accurate 70% of the time). This will decrease accuracy, but the tradeoff would be acceptable in many applications where probabilistic estimates (as opposed to discrete labels) are the key decision making tool. See Section 5.2.1 for a more thorough discussion and empirical results regarding this potential trade-off. Here we are only concerned with calibration, and leave methods for exploring the Pareto front of selective calibration and accuracy to future work.

# 4 Selective Recalibration

In order to achieve lower calibration error than existing approaches, we propose jointly optimizing a selection model and a recalibration model. Expected calibration error under both selection and recalibration is equal to

$$\text{SR-ECE}_q = \left( \mathbb{E}_{\hat{h}(f(x))} \left[ \left( \left| \mathbb{E}_{\mathcal{D}}[y^c|\hat{h}(f(x)), g(x) = 1] - \hat{h}(f(x)) \right| \right)^q \mid g(x) = 1 \right] \right)^{\frac{1}{q}}. \tag{3}$$

Taking SR-ECE$_q$ as our loss quantity of interest, our goal in selective recalibration is to solve the optimization problem:

$$\min_{g,h} \text{SR-ECE}_q \quad \text{s.t.} \quad \mathbb{E}_{\mathcal{D}}[g(x)] \geq \beta, \tag{4}$$

where $\beta$ is our desired coverage level.

There are several different ways one could approach optimizing the quantity in Eq. 4 through selection and/or recalibration. One could apply only $h$ or $g$, first train $h$ and then $g$ (or vice versa), or jointly train $g$ and $h$ (i.e., selective recalibration). In Fisch et al. (2022), only $g$ is applied; however, as our experiments will show, much of the possible reduction in calibration error comes from $h$. While $h$ can be effective alone, typical recalibrators are inexpressive, and thus may benefit from rejecting some difficult-to-fit portion of the data (as we find to be the case in experiments on several real-world datasets in Section 5). Training the models sequentially is also sub-optimal, as the benefits of selection with regards to recalibration can only be fully realized if the two models can interact in training, since fixing the first model constrains the available solutions.

Selective recalibration, where $g$ and $h$ are trained together, admits any solution available to these approaches, and can produce combinations of $g$ and $h$ that are unlikely to be found via sequential optimization (we formalize this intuition theoretically via an example in Section 6). Since Eq. 4 cannot be directly optimized, we instead follow Geifman & El-Yaniv (2019) and Fisch et al. (2022) and define a surrogate loss function $L$ including both a selective error quantity and a term to enforce the coverage constraint (weighted by $\lambda$):

$$\min_{g,h} L = L_{sel} + \lambda L_{cov}(\beta). \tag{5}$$

We describe choices for $L_{sel}$ (selection loss) and $L_{cov}$ (coverage loss) in Sections 4.1 and 4.2, along with recalibrator models in Section 4.3. Finally, we specify an inference procedure in Section 4.4, and explain how the model trained with the soft constraint in Eq. 5 is used to satisfy Eq. 4.

## 4.1 Selection Loss

In selective recalibration, the selection loss term measures the calibration of selected instances. Its general form for a batch of data $D = \{(x_i, y_i)\}_{i=1}^n$ with $n$ examples is

$$L_{sel} = \frac{l(\hat{g}, h; f, D)}{\frac{1}{n} \sum_i \hat{g}(x_i)} \tag{6}$$

where $l$ measures the loss on selected examples and the denominator scales the loss according to the proportion preserved. We consider 3 forms of $l$: S-MMCE of Fisch et al. (2022), a selective version of multi-class cross entropy, and our proposed selective top label cross entropy loss.

### 4.1.1 Maximum Mean Calibration Error

We apply the S-MMCE loss function proposed in Fisch et al. (2022) for training a selective calibration system. For a batch of training data, this loss function is defined as

$$l_{\text{S-MMCE}}(\hat{g}, h; f, D) = \left[ \frac{1}{n^2} \sum_{i,j} \left| y_i^c - \hat{h}(f(x_i)) \right|^q \left| y_j^c - \hat{h}(f(x_j)) \right|^q \hat{g}(x_i) \hat{g}(x_j) \phi\left( \hat{h}(f(x_i)), \hat{h}(f(x_j)) \right) \right]^{\frac{1}{q}} \tag{7}$$

where $\phi$ is some similarity kernel, like Laplacian. On a high level, this loss penalizes pairs of instances that have similar confidence and both are far from the true label $y^c$ (which denotes prediction correctness 0 or 1). Further details and motivation for such an objective can be found in Fisch et al. (2022).

### 4.1.2 Top Label Binary Cross Entropy

Consider a selective version of a typical multi-class cross entropy loss:

$$l_{\text{S-MCE}}(\hat{g}, h; f, D) = \frac{-1}{n} \sum_i \hat{g}(x_i) \log h(f(x_i))_{y_i} . \tag{8}$$

In the case that the model is incorrect ($y^c = 0$), this loss will penalize based on under-confidence in the ground truth class. However, our goal is calibration according to the *predicted* class. Thus we propose a loss function for training a selective recalibration model based on the typical approach to optimizing a smooth recalibration model, Selective Top Label Binary Cross Entropy (S-TLBCE):

$$l_{\text{S-TLBCE}}(\hat{g}, h; f, D) = \frac{-1}{n} \sum_i \hat{g}(x_i) \Big[ y_i^c \log \hat{h}(f(x_i)) + (1 - y_i^c) \log(1 - \hat{h}(f(x_i))) \Big]. \tag{9}$$

In contrast to S-MCE, in the case of an incorrect prediction S-TLBCE penalizes based on over-confidence in the predicted label. This aligns with the established notion of top-label calibration error, as well as the typical Platt or temperature scaling objectives, and makes this a natural loss function for training a selective recalibration model. We compare S-TLBCE and S-MCE empirically in our experiments, and note that in the binary case these losses are the same.

## 4.2 Coverage Loss

When the goal of selection is improving accuracy, there exists an ordering under $\hat{g}$ that is optimal for any choice of $\beta$, namely that where $\hat{g}$ is greater for all correctly labeled examples than it is for any incorrectly labeled example. Accordingly, coverage losses used to train these systems will often only enforce that *no more than* $\beta$ proportion is to be rejected. Unlike selective accuracy, selective calibration is not monotonic with respect to individual examples and a mismatch in coverage between training and deployment may hurt test performance. Thus in selective recalibration we assume the user aims to reject exactly $\beta$ proportion of the data, and employ a coverage loss that targets a specific $\beta$,

$$L_{cov}(\beta) = \Big(\beta - \frac{1}{n} \sum_i \hat{g}(x_i)\Big)^2. \tag{10}$$

Such a loss will be an asymptotically consistent estimator of $(\beta - \mathbb{E}[\hat{g}(x)])^2$. Alternatively, Fisch et al. (2022) use a logarithmic regularization approach for enforcing the coverage constraint without a specific target $\beta$, computing cross entropy between the output of $\hat{g}$ and a target vector of all ones. However, we found this approach to be unstable and sensitive to the choice of $\lambda$ in initial experiments, while our coverage loss enabled stable training at any sufficiently large choice of $\lambda$, similar to the findings of Geifman & El-Yaniv (2019).

## 4.3 Recalibration Models

We consider two differentiable and popular calibration models, Platt scaling and temperature scaling, both of which attempt to fit a function between model confidence and output correctness. The main difference between the models is that Platt scaling works in the output probability space, whereas temperature scaling is applied to model logits before a softmax is taken. A Platt recalibrator (Platt, 1999) produces output according to

$$h^{\text{Platt}}(f(x)) = \frac{1}{1 + \exp(wf(x) + b)} \tag{11}$$

where $w, b$ are learnable scalar parameters. On the other hand, a temperature scaling model (Guo et al., 2017) produces output according to

$$h^{\text{Temp}}(f_s(x)) = \text{softmax}\left(\frac{f_s(x)}{T}\right) \tag{12}$$

where $f_s(x)$ is the vector of logits (unnormalized scores) produced by $f$ and $T$ is the single learned (scalar) parameter. Both models are typically trained with a binary cross-entropy loss, where the labels 0 and 1 denote whether an instance is correctly classified.

## 4.4 Inference

Once we have trained $\hat{g}$ and $h$, we can flexibly account for $\beta$ by selecting a threshold $\tau$ on unlabeled test data (or some other representative tuning set) such that $\mathbb{E}[\mathbf{1}\{\hat{g}(X) \geq \tau\}] = \beta$. The model $g$ is then simply $g(x) := \mathbf{1}\{\hat{g}(x) \geq \tau\}$.

### 4.4.1 High Probability Coverage Guarantees

Since $\mathbf{1}\{\hat{g}(x) \geq \tau\}$ is a random variable with a Bernoulli distribution, we may also apply the Hoeffding bound (Hoeffding, 1963) to guarantee that with high probability empirical target coverage $\hat{\beta}$ (the proportion of the target distribution where $\hat{g}(x) \geq \tau$) will be in some range. Given a set $\mathcal{V}$ of $n_u$ i.i.d. unlabeled examples from the target distribution, we denote empirical coverage on $\mathcal{V}$ as $\tilde{\beta}$. With probability at least $1 - \delta$, $\hat{\beta}$ will be in the range $[\tilde{\beta} - \epsilon, \tilde{\beta} + \epsilon]$, where $\epsilon = \sqrt{\frac{\log(\frac{2}{\delta})}{2n_u}}$. For some critical coverage level $\beta$, $\tau$ can be decreased until $\tilde{\beta} - \epsilon \geq \beta$.

## 5 Experiments

In this section we examine the performance of selective recalibration and baselines when applied to models pre-trained on real-world datasets and applied to a target distribution possibly shifted from the training distribution. In Section 5.1 we investigate whether, given a small validation set of labeled examples drawn i.i.d. from the target distribution, joint optimization consistently leads to a lower empirical selective calibration error than selection or recalibration alone or sequential optimization. Subsequently, in Section 5.2 we study multiple out-of-distribution prediction tasks and the ability of a single system to provide decreasing selective calibration error across a range of coverage levels when faced with a further distribution shift from validation data to test data. We also analyze the trade-off between selective calibration error and accuracy in this setting.

Since we are introducing the objective of selective recalibration here, we focus on high-level design decisions, in particular the choice of selection and recalibration method, loss function, and optimization procedure (joint vs. sequential). For selective recalibration models, the input to $g$ is the feature embedding. Temperature scaling is used for multi-class examples and Platt scaling is applied in the binary cases (following Guo et al. (2017) and initial results on validation data). Calibration error is measured using $\text{ECE}_1$ and $\text{ECE}_2$ with equal mass binning. For the selection loss, we use $l_{\text{S-TLBCE}}$ and $l_{\text{S-MMCE}}$ for binary tasks, and include a selective version of typical multi-class cross-entropy ($l_{\text{S-MCE}}$) for multi-class tasks. Pre-training is performed where $h$ is optimized first in order to reasonably initialize the calibrator parameters before beginning to train $g$. Models are trained both with $h$ fixed after this pre-training (denoted as "sequential" in results) and when it is jointly optimized throughout training (denoted as "joint" in results).

Our selection baselines include confidence-based rejection ("Confidence") and multiple out-of-distribution (OOD) detection methods ("Iso. Forest", "One-class SVM"), common techniques when rejecting to improve accuracy. The confidence baseline rejects examples with the smallest $\hat{f}(x)$ (or $\hat{h}(f(x))$), while the OOD methods are measured in the embedding space of the pre-trained model. All selection baselines are applied to the recalibrated model in order to make the strongest comparison. We make further comparisons to recalibration baselines, including the previously described temperature and Platt scaling as well as binning methods like histogram binning and Platt binning. See Appendix A for more experiment details including calibration error measurement and baseline implementations.

### 5.1 Selective Recalibration with i.i.d. Data

First, we test whether selective recalibration consistently produces low ECE in a setting where there is a validation set of labeled training data available from the same distribution as test data using outputs of pretrained models on the Camelyon17 and ImageNet datasets. Camelyon17 (Bandi et al., 2018) is a task where the input $x$ is a 96x96 patch of a whole-slide image of a lymph node section from a patient with potentially metastatic breast cancer and the label $y$ is whether the patch contains a tumor. Selection and recalibration models are trained with 1000 samples, and we apply a Platt scaling $h$ since the task is binary. ImageNet is a well-known large scale image classification dataset, where we use 2000 samples from a supervised ResNet34 model for training selection and recalibration models, and temperature scaling $h$ since the task is multi-class. Our soft selector $\hat{g}$ is a shallow fully-connected network (2 hidden layers with dimension 128), and we report selective calibration error for coverage level $\beta \in \{0.75, 0.8, 0.85, 0.9\}$. Full experiment details, including model specifications and training parameters, can be found in Appendix A.

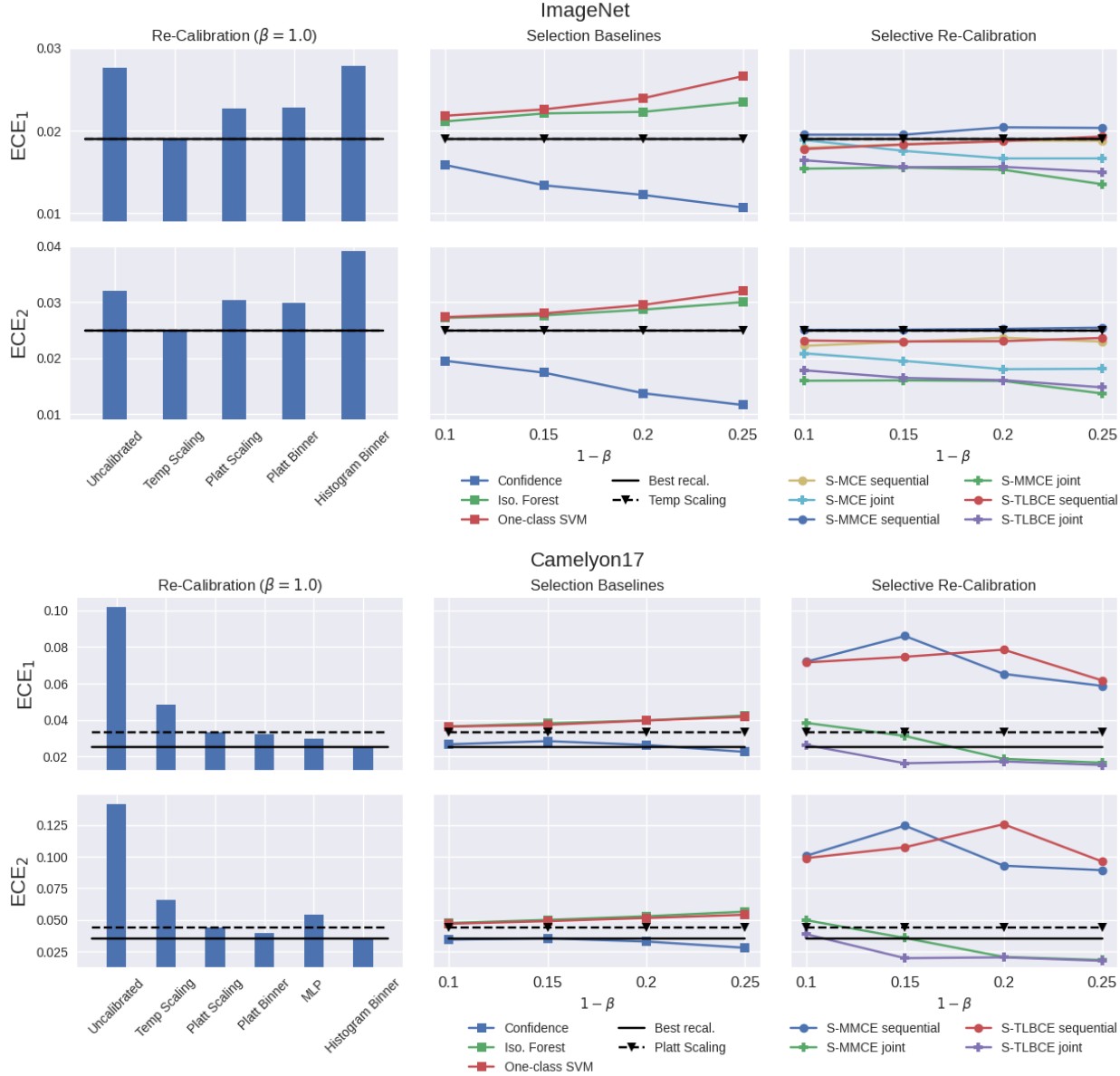

Figure 2: Selective calibration error on ImageNet and Camelyon17 for coverage level $\beta \in \{0.75, 0.8, 0.85, 0.9\}$. **Left**: Various recalibration methods are trained using labeled validation data. **Middle**: Selection baselines including confidence-based rejection and various OOD measures. **Right**: Selective recalibration with different loss functions.

Results are displayed in Figure 2. They show that by jointly optimizing the selector and recalibration models, we are able to achieve improved ECE at the given coverage level $\beta$ compared to first training $h$ and then $g$. We also find selective recalibration achieves the lowest ECE in almost every case in comparison to recalibration alone. While the confidence-based selection strategy performs well in these experiments, this is not a good approach to selective calibration in general, as this is a heuristic strategy and may fail in cases where a model's confident predictions are in fact poorly calibrated (see Section 5.2 for examples). In addition, the S-TLBCE loss shows more consistent performance than S-MMCE, as it reduces ECE in every case, whereas training with S-MMCE increase calibration error in some cases.

To be sure that the lower calibration error as compared to recalibration is not because of the extra parameters in the selector model, we also produce results for a recalibration model with the same architecture as our selector. Once again the input is the feature embedding, and the model $h$ has 2 hidden layers with dimension 128. Results for Camelyon17 are included in Figure 2; for clarity of presentation of ImageNet results, we omit the MLP recalibrator results from the plot as they were an order of magnitude worse than all other methods (ECE$_1$ 0.26, ECE$_2$ 0.30). In neither case does this baseline reach the performance of the best low-dimensional recalibration model.

## 5.2 Selective Re-Calibration under Distribution Shift

In this experiment, we study the various methods applied to genetic perturbation classification with RxRx1, as well as zero-shot image classification with CLIP and CIFAR-100-C. RxRx1 (Taylor et al., 2019) is a task where the input $x$ is a 3-channel image of cells obtained by fluorescent microscopy, the label $y$ indicates which of 1,139 genetic treatments (including no treatment) the cells received, and there is a domain shift across the batch in which the imaging experiment was run. CIFAR-100 is a well-known image classification dataset, and we perform zero-shot image classification with CLIP. We follow the setting of Fisch et al. (2022) where the test data is drawn from a shifted distribution with respect to the validation set and the goal is not to target a specific $\beta$, but rather to train a selector that works across a range of coverage levels. In the case of RxRx1 the strong batch processing effect leads to a 9% difference in pretrained model accuracy between validation (18%) and test (27%) output, and we also apply a selective recalibration model trained on validation output from zero-shot CLIP and CIFAR-100 to test examples drawn from the OOD CIFAR-100-C test set. Our validation sets have 1000 (RxRx1) or 2000 (CIFAR-100) examples, $\hat{g}$ is a small network with 1 hidden layer of dimension 64, and we set $\beta = 0.5$ when training the models. For our results we report the area under the curve (AUC) for the coverage vs. error curve, a typical metric in selective classification (Geifman & El-Yaniv, 2017; Fisch et al., 2022) that reflects how a model can reduce the error on average at different levels of $\beta$. We measure AUC in the range $\beta = [0.5, 1.0]$, with measurements taken at intervals of 0.05 (i.e., $\beta \in [0.5, 0.55, 0.6, ..., 0.95, 1.0]$). Additionally, to induce robustness to the distribution shift we noise the selector/recalibrator input. See Appendix A for full specifications.

Table 1: RxRx1 and CIFAR-100-C AUC in the range $\beta = [0.5, 1.0]$.

| Selection | Opt. of $h, g$ | RxRx1 | | | CIFAR-100-C | | |
|---|---|---|---|---|---|---|---|
| | | ECE$_1$ | ECE$_2$ | Acc. | ECE$_1$ | ECE$_2$ | Acc. |
| Confidence | - | 0.071 | 0.081 | **0.353** | 0.048 | 0.054 | **0.553** |
| One-class SVM | - | 0.058 | 0.077 | 0.227 | 0.044 | 0.051 | 0.388 |
| Iso. Forest | - | 0.048 | 0.061 | 0.221 | 0.044 | 0.051 | 0.379 |
| S-MCE | sequential | 0.059 | 0.075 | 0.250 | 0.033 | 0.041 | 0.499 |
| | joint | 0.057 | 0.073 | 0.249 | 0.060 | 0.068 | 0.484 |
| S-MMCE | sequential | **0.036** | **0.045** | 0.218 | 0.030 | 0.037 | 0.503 |
| | joint | **0.036** | **0.045** | 0.218 | 0.043 | 0.051 | 0.489 |
| S-TLBCE | sequential | **0.036** | **0.045** | 0.219 | 0.030 | 0.037 | 0.507 |
| | joint | 0.039 | 0.049 | 0.218 | **0.026** | **0.032** | 0.500 |
| Recalibration | Temp. Scale | 0.055 | 0.070 | 0.274 | 0.041 | 0.047 | 0.438 |
| ($\beta = 1.0$) | None | 0.308 | 0.331 | 0.274 | 0.071 | 0.079 | 0.438 |

Results are shown in Table 1. First, these results highlight that even in this OOD setting, the selection-only approach of Fisch et al. (2022) is not enough and recalibration is a key ingredient in improving selective calibration error. Fixing $h$ and then training $g$ performs better than joint optimization for RxRx1, likely because the distribution shift significantly changed the optimal temperature for the region of the feature space where $g(x) = 1$. Joint optimization performs best for CIFAR-100-C, and does still significantly improve ECE on RxRx1, although it's outperformed by fixing $h$ first in that case. The confidence baseline performs quite poorly on both experiments and according to both metrics, significantly increasing selective calibration error in all cases.

### 5.2.1 Trade-offs Between Calibration Error and Accuracy

While accurate probabilistic output is the only concern in some domains and should be of at least some concern in most domains, discrete label accuracy can also be important in some circumstances. Table 1 shows accuracy results under selection, and Figure 3 shows the selective accuracy curve and confidence histogram for our selective recalibration model trained with S-TLBCE for RxRx1 and CIFAR-100 (and applied to shifted distributions). Together, these results illustrate that under different data and prediction distributions, selective recalibration may increase or decrease accuracy. For RxRx1, the model tends to reject examples with higher confidence, which also tend to be more accurate. Thus, while ECE@$\beta$ may improve with respect to the full dataset, selective accuracy at $\beta$ is worse. On the other hand, for CIFAR-100-C, the model tends to reject examples with lower confidence, which also tend to be less accurate. Accordingly, both ECE@$\beta$ and selective accuracy at $\beta$ improve with respect to the full dataset.

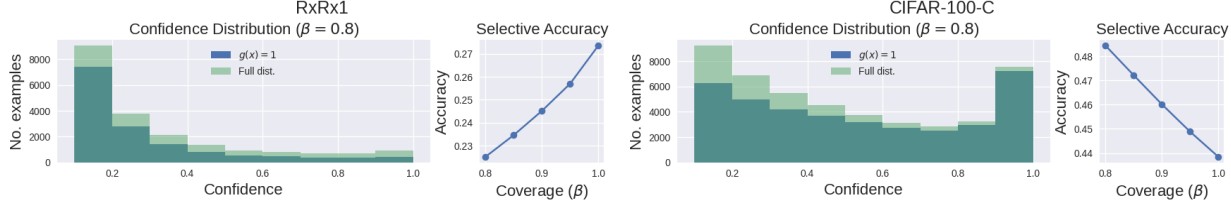

Figure 3: Plots illustrating 1) distribution of confidence among the full distribution and those examples accepted for prediction (i.e., where $g(x) = 1$) at coverage level $\beta = 0.8$ and 2) selective accuracy in the range $\beta = [0.8, 1.0]$.

## 6 Theoretical Analysis

To build a deeper understanding of selective recalibration (and its alternatives), we consider a theoretical situation where a pre-trained model is applied to a target distribution different from the distribution on which it was trained, mirroring both our experimental setting and a common challenge in real-world deployments. We show that with either selection or recalibration alone there will still be calibration error, while selective recalibration can achieve ECE = 0. We also show that joint optimization of $g$ and $h$, as opposed to sequentially optimizing each model, is necessary to achieve zero calibration error.

### 6.1 Setup

We consider a setting with two classes, and without loss of generality we set $y \in \{-1, 1\}$. We are given a classifier pre-trained on a mixture model, a typical way to view the distribution of objects in images (Zhu et al., 2014; Thulasidasan et al., 2019). The pre-trained classifier is then applied to a target distribution containing a portion of outliers from each class unseen during training. Our specific choices of classifier and training and target distributions are chosen for ease of interpretation and analysis; however, the intuitions built can be applied more generally, for example to neural network classifiers, which are too complex for such analysis but are often faced with outlier data on which calibration is poor (Ovadia et al., 2019).

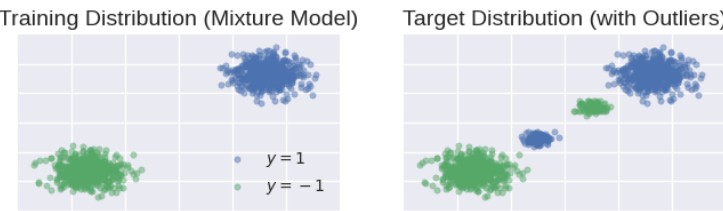

Figure 4: A classifier pre-trained on a mixture model is applied to a target distribution with outliers.

### 6.1.1 Data Generation Model

**Definition 1 (Target Distribution)** *The target distribution is defined as a $(\theta^*, \sigma, \alpha)$-perturbed mixture model over $(x, y) \in \mathbb{R}^p \times \{1, -1\}$: $x \mid y, z \sim zJ_1 + (1 - z)J_2$, where $y$ follows the Bernoulli distribution $\mathbb{P}(y = 1) = \mathbb{P}(y = -1) = 1/2$, $z$ follows a Bernoulli distribution $\mathbb{P}(z = 1) = \beta$, and $z$ is independent of $y$.*

Our data model considers a mixture of two distributions with disjoint and bounded supports, $J_1$ and $J_2$, where $J_2$ is considered to be an outlier distribution. Specifically, for $y \in \{-1, 1\}$, $J_1$ is supported in the balls with centers $y\theta^*$ and radius $r_1$, $J_2$ is supported in the balls with centers $-y\alpha\theta^*$ and radius $r_2$, and both $J_1$ and $J_2$ have standard deviation $\sigma$. See Figure 4 for an illustration of our data models, and Appendix C.1 for a full definition of the distribution.

### 6.1.2 Classifier Algorithm

Recall that in our setting $f$ is a pre-trained model, where the training distribution is unknown and we only have samples from some different target distribution. We follow this setting in our theory by considering a (possibly biased) estimator $\hat{\theta}$ of $\theta^*$, which is the output of a training algorithm $\mathscr{A}(S^{tr})$ that takes the i.i.d. training data set $S^{tr} = \{(x_i^{tr}, y_i^{tr})\}_{i=1}^m$ as input. The distribution from which $S^{tr}$ is drawn is ***different*** from the target distribution from which we have data to train the selection and recalibration models. We only impose one assumption on the model $\hat{\theta}$: that $\mathscr{A}$ outputs a $\hat{\theta}$ that will converge to $\theta_0$ if training data is abundant enough and $\theta_0$ should be aligned with $\theta^*$ with respect to direction (see Assumption 3, Appendix C.2 for formal statement). For ease of analysis and explanation, we consider a simple classifier defined by $\hat{\theta} = \sum_{i=1}^m x_i^{tr} y_i^{tr}/m$ when the training distribution is set to be an unperturbed Gaussian mixture $x^{tr} | y^{tr} \sim \mathcal{N}(y^{tr} \cdot \theta^*, \sigma^2 I)$ and $y^{tr}$ follows a Bernoulli distribution $\mathbb{P}(y^{tr} = 1) = 1/2$.[1] This form of $\hat{\theta}$ is closely related to Fisher's rule in linear discriminant analysis for Gaussian mixtures (see Appendix C.2.1 for further discussion).

Having obtained $\hat{\theta}$, our pretrained classifier aligns with the typical notion of a softmax response in neural networks. We first obtain the confidence vector $f(x) = (f_1(x), f_{-1}(x))^\top$, where

$$f_{-1}(x) = \frac{1}{e^{2\hat{\theta}^\top x} + 1}, \quad f_1(x) = \frac{e^{2\hat{\theta}^\top x}}{e^{2\hat{\theta}^\top x} + 1}. \tag{13}$$

and then output $\hat{y} = \arg\max_{k \in \{-1, 1\}} f_k(x)$. For $k \in \{-1, 1\}$, the confidence score $f_k(x)$ represents an estimator of $\mathbb{P}(y = k|x)$ and the final classifier is equivalent to $\hat{y} = \operatorname{sgn}(\hat{\theta}^\top x)$.

### 6.2 Main Theoretical Results

Having established our data and classification models, we now analyze why selective recalibration (i.e., joint training of $g$ and $h$) can outperform recalibration and selection performed alone or sequentially. To measure calibration error, we consider $\text{ECE}_q$ with $q = 1$ (and drop the subscript $q$ for notational simplicity below). For the clarity of theorem statements and proofs, we will restate definitions of calibration error to make them explicitly dependent on selection model $g$ and temperature $T$ and tailored for the binary case we are studying. We want to emphasize that we are ***not** introducing new concepts*, but instead offering different surface forms of the same quantities introduced earlier. First, we notice that under our data generating model and pretrained classifier, ECE can be expressed as

$$\text{ECE} = \mathbb{E}_{\hat{\theta}^\top x}\left[\left|\mathbb{P}[y = 1 \mid \hat{\theta}^\top x = v] - \frac{1}{1 + e^{-2v}}\right|\right].$$

By studying such population quantities, our analysis is not dependent on any binning-methods that are commonly used in empirically calculating expected calibration errors.

---

[1]The in-distribution case also works under our data generation model.

### 6.2.1 Selective Recalibration v.s. Recalibration or Selection

We study the following ECE quantities according to our data model for recalibration alone (R-ECE), selection alone (S-ECE), and selective recalibration (SR-ECE). For recalibration, we focus on studying the popular temperature scaling model, although the analysis is nearly identical for Platt scaling.

$$\text{R-ECE}(T) = \mathbb{E}_{\hat{\theta}^\top x}\left[\left|\mathbb{P}[y = 1 \mid \frac{\hat{\theta}^\top x}{T} = v] - \frac{1}{1 + e^{-2v}}\right|\right].$$

$$\text{S-ECE}(g) := \mathbb{E}_{\hat{\theta}^\top x}\left[\left|\mathbb{P}[y = 1 \mid \hat{\theta}^\top x = v, g(x) = 1] - \frac{1}{1 + e^{-2v}}\right| \mid g(x) = 1\right].$$

$$\text{SR-ECE}(g, T) := \mathbb{E}_{\hat{\theta}^\top x}\left[\left|\mathbb{P}[y = 1 \mid \frac{\hat{\theta}^\top x}{T} = v, g(x) = 1] - \frac{1}{1 + e^{-2v}}\right| \mid g(x) = 1\right].$$

Our first theorem proves that under our data generation model, S-ECE and R-ECE can never reach 0, but SR-ECE can reach 0 by choosing appropriate $g$ and $T$.

**Theorem 1** *Under Assumption 3, for any $\delta \in (0, 1)$ and $\hat{\theta}$ output by $\mathscr{A}$, there exist thresholds $M \in \mathbb{N}^+$ and $\tau > 0$ such that if $\max\{r_1, r_2, \sigma, \|\theta^*\|\} < \tau$ and $m > M$, there exists a positive lower bound $L$, with probability at least $1 - \delta$ over $S^{tr}$*

$$\min\left\{\min_{g:\mathbb{E}[g(x)]\geq\beta} S\text{-}ECE(g), \quad \min_{T\in\mathbb{R}} R\text{-}ECE(T)\right\} > L.$$

*However, there exists $T_0$ and $g_0$ satisfying $\mathbb{E}[g_0(x)] \geq \beta$, such that $SR\text{-}ECE(g_0, T_0) = 0$.*

**Intuition and interpretation.** Here we give some intuition for understanding our results. Under our perturbed mixture model, R-ECE is calculated as

$$\text{R-ECE}(T) = \mathbb{E}_{v=\hat{\theta}^\top x}\left|\frac{\mathbf{1}\{v \in \mathcal{A}\}}{1 + \exp\left(\frac{-2\hat{\theta}^\top\theta^*}{\sigma^2\|\hat{\theta}\|^2} \cdot v\right)} + \frac{\mathbf{1}\{v \in \mathcal{B}\}}{1 + \exp\left(\frac{2\alpha\hat{\theta}^\top\theta^*}{\sigma^2\|\hat{\theta}\|^2} \cdot v\right)} - \frac{1}{e^{-2v/T} + 1}\right|$$

for disjoint sets $\mathcal{A}$ and $\mathcal{B}$, which correspond to points on the support of $J_1$ and $J_2$ respectively. In order to achieve zero R-ECE, when $v \in \mathcal{A}$, we need $T = \hat{\theta}^\top\theta^*/(\sigma^2\|\hat{\theta}\|^2)$. However, for $v \in \mathcal{B}$ we need $T = -\alpha\hat{\theta}^\top\theta^*/(\sigma^2\|\hat{\theta}\|^2)$. These clearly cannot be achieved simultaneously. Thus the presence of the outlier data makes it impossible for the recalibration model to properly calibrate the confidence for the whole population. A similar expression can be obtained for S-ECE. As long as $\hat{\theta}^\top\theta^*/(\sigma^2\|\hat{\theta}\|^2)$ and $-\alpha\hat{\theta}^\top\theta^*/(\sigma^2\|\hat{\theta}\|^2)$ are far from 1 (i.e., miscalibration exists), no choice of $g$ can reach zero S-ECE. In other words, no selection rule alone can lead to calibrated predictions, since no subset of the data is calibrated under the pre-trained classifier. However, by setting $g_0(x) = 0$ for all $x \in \mathcal{B}$ and $g_0(x) = 1$ otherwise, and choosing $T_0 = \hat{\theta}^\top\theta^*/(\sigma^2\|\hat{\theta}\|^2)$, SR-ECE $= 0$. Thus we can conclude that achieving ECE $= 0$ on eligible predictions is only possible under selective recalibration, while selection or recalibration alone induce positive ECE. See Appendix C for more details and analysis.

### 6.2.2 Joint Learning versus Sequential Learning

We can further demonstrate that jointly learning a selection model $g$ and temperature scaling parameter $T$ can outperform sequential learning of $g$ and $T$. Let us first denote $\tilde{g} := \arg\min_g \text{S-ECE}(g)$ such that $\mathbb{E}[\tilde{g}(x)] \geq \beta$ and $\tilde{T} := \arg\min_T \text{R-ECE}(T)$. We denote two types of expected calibration error under sequential learning of $g$ and $T$, depending on which is optimized first.

$$\text{ECE}^{R\rightarrow S} := \min_{g:\mathbb{E}[g(x)]\geq\beta} \text{S-ECE}(g, \tilde{T});$$

$$\text{ECE}^{S\rightarrow R} := \min_{T\in\mathbb{R}} \text{R-ECE}(\tilde{g}, T).$$

Our second theorem shows these two types of expected calibration error for sequential learning are lower bounded, while jointly learning $g, T$ can reach zero calibration error.

**Theorem 2** *Under Assumption 3, if $\beta > 2(1 - \beta)$, for any $\delta \in (0, 1)$ and $\hat{\theta}$ output by $\mathscr{A}$, there exist thresholds $M \in \mathbb{N}^+$ and $\tau_2 > \tau_1 > 0$: if $\max\{r_1, r_2, \sigma\} < \tau_2$, $\tau_1 < \sigma$, and $m > M$, then there exists a positive lower bound $L$, with probability at least $1 - \delta$ over $S^{tr}$*

$$\min\left\{ ECE^{R \to S}, ECE^{S \to R} \right\} > L.$$

*However, there exists $T_0$ and $g_0$ satisfying $\mathbb{E}[g_0(x)] \geq \beta$, such that SR-ECE$(g_0, T_0) = 0$.*

**Intuition and interpretation.** If we first optimize the temperature scaling model to obtain $\tilde{T}$, $\tilde{T}$ will not be equal to $\hat{\theta}^\top \theta^* / (\sigma^2 \|\hat{\theta}\|^2)$. Then, when applying selection, there exists no $g$ that can reach 0 calibration error since the temperature is not optimal for data in $\mathcal{A}$ or $\mathcal{B}$. On the other hand, if we first optimize the selection model and obtain $\tilde{g}$, $\tilde{g}$ will reject points in $\mathcal{A}$ instead of those in $\mathcal{B}$ because points in $\mathcal{A}$ incur higher calibration error, and thus data from both $\mathcal{A}$ and $\mathcal{B}$ will be selected (i.e., not rejected). In that case, temperature scaling not will be able to push calibration error to zero because, similar to the case in the earlier R-ECE analysis, the calibration error in $\mathcal{A}$ and $\mathcal{B}$ cannot reach 0 simultaneously using a single temperature scaling model. On the other hand, the optimal jointly-learned solution yields a set of predictions with zero expected calibration error.

# 7 Conclusion

We have shown both empirically and theoretically that combining selection and recalibration is a potent strategy for producing a set of well-calibrated predictions. Eight pairs of distribution and $\beta$ were tested when i.i.d. validation data is available; selective recalibration with our proposed S-TLBCE loss function outperforms every single recalibrator tested in 7 cases, and always reduces S-ECE with respect to the calibrator employed by the selective recalibration model itself. Taken together, these results show that while many popular recalibration functions are quite effective at reducing calibration error, they can often be better fit to the data when given the opportunity to ignore a small portion of difficult examples. Thus, in domains where calibrated confidence is critical to decision making, selective recalibration is a practical and lightweight strategy for improving outcomes downstream of deep learning model predictions.

**Broader Impact Statement**

While the goal of our method is to foster better outcomes in settings of societal importance like medical diagnosis, as mentioned in Section 2, selective classification may create disparities among protected groups. Future work on selective recalibration could focus on analyzing and mitigating any unequal effects of the algorithm.

**Acknowledgments**

We thank Mike Mozer and Jasper Snoek for their very helpful feedback on this work. This research obtained support by the funds provided by the National Science Foundation and by DoD OUSD (R&E) under Cooperative Agreement PHY-2229929 (ARNI: The NSF AI Institute for Artificial and Natural Intelligence). JCS gratefully acknowledges financial support from the Schmidt DataX Fund at Princeton University made possible through a major gift from the Schmidt Futures Foundation. TP gratefully acknowledges support by NSF AF:Medium 2212136 and by the Simons Collaboration on the Theory of Algorithm Fairness grant.

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

# Appendix

# A   Additional Experiment Details

In training we follow Fisch et al. (2022) and drop the denominator in $L_{sel}$, as the coverage loss suffices to keep $\hat{g}$ from collapsing to 0. Recalibration model code is taken from the accompanying code releases from Guo et al. (2017)[2] (Temperature Scaling) and Kumar et al. (2019)[3] (Platt Scaling, Histogram Binning, Platt Binning).

## A.1   Calibration Measures

We calculate $\text{ECE}_q$ for $q \in \{1, 2\}$ using the python library released by Kumar et al. (2019) [4]. $\text{ECE}_q$ is calculated as:

$$\text{ECE}_q = \left( \frac{1}{|B|} \sum_{j=1}^{|B|} \left( \frac{\sum_{i \in B_j} \mathbf{1}\{y_i = \hat{y}_i\}}{|B_j|} - \frac{\sum_{i \in B_j} \hat{f}(x_i)}{|B_j|} \right)^q \right)^{\frac{1}{q}} \tag{14}$$

where $B = B_1, ..., B_m$ are a set of $m$ equal-mass prediction bins, and predictions are sorted and binned based on their maximum confidence $\hat{f}(x)$. We set $m = 15$.

## A.2   Baselines

Next we describe how baseline methods are implemented. Our descriptions are based on creating an ordering of the test set such that at a given coverage level $\beta$, a $1 - \beta$ proportion of examples from the end of the ordering are rejected.

### A.2.1   Confidence-Based Rejection

Confidence based rejection is performed by ordering instances in a decreasing order based on $\hat{f}(x)$, the maximum confidence the model has in any class for that example.

### A.2.2   Out of Distribution Scores

The sklearn python library (Pedregosa et al., 2011) is used to produce the One-Class SVM and Isolation Forest models. Anomaly scores are oriented such that more typical datapoints are given higher scores; instances are ranked in a decreasing order.

## A.3   In-distribution Experiments

Our selector $g$ is a shallow fully-connected network with 2 hidden layers of dimension 128 and ReLU activations.

### A.3.1   Camelyon17

Camelyon17 (Bandi et al., 2018) is a task where the input $x$ is a 96x96 patch of a whole-slide image of a lymph node section from a patient with potentially metastatic breast cancer, the label $y$ is whether the patch contains a tumor, and the domain $d$ specifies which of 5 hospitals the patch was from. We pre-train a DenseNet-121 model on the Camelyon17 train set using the code from Koh et al. (2021)[5]. The validation set has 34,904 examples and accuracy of 91%, while the test set has 84,054 examples, and accuracy of 83%. Our selector $g$ is trained with a learning rate of 0.0005, the coverage loss weight $\lambda$ is set to 32 (following (Geifman & El-Yaniv, 2019)), and the model is trained with 1000 samples for 1000 epochs with a batch size of 100.

---

[2]https://github.com/gpleiss/temperature_scaling
[3]https://github.com/p-lambda/verified_calibration
[4]https://github.com/p-lambda/verified_calibration
[5]https://github.com/p-lambda/wilds

### A.3.2 ImageNet

ImageNet is a large scale image classification dataset. We extract the features, scores, and labels from the 50,000 ImageNet validation samples using a pre-trained ResNet34 model from the torchvision library. Our selector $g$ is trained with a learning rate of 0.00001, the coverage loss weight $\lambda$ is set to 32 (following (Geifman & El-Yaniv, 2019)), and the model is trained with 2000 samples for 1000 epochs with a batch size of 200.

### A.4 Out-of-distribution Experiments

Our selector $g$ is a shallow fully-connected network (1 hidden layer with dimension 64 and ReLu activation) trained with a learning rate of 0.0001, the coverage loss weight $\lambda$ is set to 8, and the model is trained for 50 epochs (to avoid overfitting since this is an OOD setting) with a batch size of 256.

### A.4.1 RxRx1

RxRx1 (Taylor et al., 2019) is a task where the input $x$ is a 3-channel image of cells obtained by fluorescent microscopy, the label $y$ indicates which of the 1,139 genetic treatments (including no treatment) the cells received, and the domain $d$ specifies the batch in which the imaging experiment was run. The validation set has 9,854 examples and accuracy of 18%, while the test set has 34,432 examples, and accuracy of 27%. 1000 samples are drawn for model training. Gaussian noise with mean 0 and standard deviation 1 is added to training examples in order to promote robustness.

### A.4.2 CIFAR-100

CIFAR-100 is a well-known image classification dataset, and we perform zero-shot image classification with CLIP. We draw 2000 samples for model training, and test on 50,000 examples drawn from the 750,000 examples in CIFAR-100-c. Data augmentation in training is performed using AugMix (Hendrycks et al., 2020) with a severity level of 3 and a mixture width of 3.

## B  Additional Experiment Results

### B.1  Brier Score Results

While our focus in this work is Expected Calibration Error, for completeness we also report results with respect to Brier Score. Figure 5 shows Brier score results for the experiments in Section 5.1. Selective recalibration reduces Brier score in both experiments and outperforms recalibration. The OOD selection baselines perform well, although they show increasing error as more data is rejected, illustrating their poor fit for the task. Further, Brier score results for the experiments in Section 5.2 are included in Table 2. Selective recalibration reduces error, and confidence-based rejection increases error, which is surprising since Brier score favors predictions with confidence near 1.

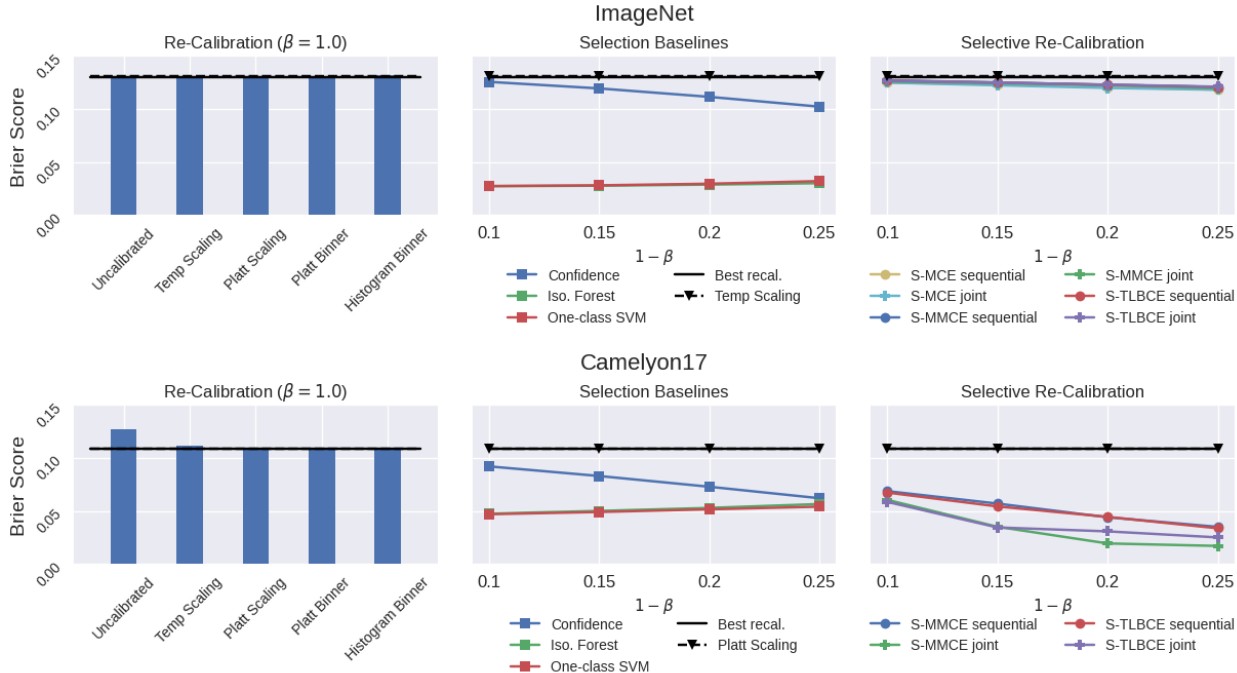

Figure 5: Selective calibration error on ImageNet and Camelyon17 for coverage level $\beta \in \{0.75, 0.8, 0.85, 0.9\}$. **Left**: Various re-calibration methods are trained using labeled validation data. **Middle**: Selection baselines including confidence-based rejection and various OOD measures. **Right**: Selective re-calibration with different loss functions.

Table 2: RxRx1 and CIFAR-100-C AUC in the range $\beta = [0.5, 1.0]$.

| Selection | Opt. of $h, g$ | RxRx1 | CIFAR-100-C |
|---|---|---|---|
| Confidence | - | 0.169 | 0.180 |
| One-class SVM | - | 0.077 | **0.051** |
| Iso. Forest | - | **0.061** | **0.051** |
| S-MCE | sequential | 0.138 | 0.166 |
| S-MCE | joint | 0.138 | 0.166 |
| S-MMCE | sequential | 0.126 | 0.165 |
| S-MMCE | joint | 0.126 | 0.164 |
| S-TLBCE | sequential | 0.126 | 0.166 |
| S-TLBCE | joint | 0.126 | 0.165 |
| Recalibration | Temp. Scale | 0.140 | 0.164 |
| ($\beta = 1.0$) | None | 0.252 | 0.170 |

# C  Theory: Additional Details and Proofs

## C.1  Details on Data Generation Model

**Definition 2 (Formal version of definition 1)** *For $\theta^* \in \mathbb{R}^p$, a $(\theta^*, \sigma, \alpha, r_1, r_2)$-perturbed truncated-Gaussian model is defined as the following distribution over $(x, y) \in \mathbb{R}^p \times \{1, -1\}$:*

$$x \mid y \sim z J_1 + (1 - z) J_2.$$

*Here, $J_1$ and $J_2$ are two truncated Guassian distributions, i.e.,*

$$J_1 \sim \rho_1 \mathcal{N}(y \cdot \theta^*, \sigma^2 I) \mathbf{1}\{x \in \mathbb{B}(\theta^*, r_1) \cup \mathbb{B}(-\theta^*, r_1)\},$$

$$J_2 \sim \rho_2 \mathcal{N}(-y \cdot \alpha\theta^*, \sigma^2 I) \mathbf{1}\{x \in \mathbb{B}(\alpha \cdot \theta^*, r_2) \cup \mathbb{B}(-(\alpha \cdot \theta^*), r_2)\}$$

*where $\rho_1, \rho_2$ are normalization coefficients to make $J_1$ and $J_2$ properly defined; $y$ follows the Bernoulli distribution $\mathbb{P}(y = 1) = \mathbb{P}(y = -1) = 1/2$; and $z$ follows a Bernoulli distribution $\mathbb{P}(z = 1) = \beta$.*

*For simplicity, throughout this paper, we set $\rho_1 = \rho_2$ and this is always achievable by setting $r_1/r_2$ appropriately. We also set $\alpha \in (0, 1/2)$.*

## C.2   Details on $\hat{\theta}$

Recall that we consider the $\hat{\theta}$ that is the output of a training algorithm $\mathscr{A}(S^{tr})$ that takes the i.i.d. training data set $S^{tr} = \{(x_i^{tr}, y_i^{tr})\}_{i=1}^m$ as input. We imposed the following assumption on $\hat{\theta}$.

**Assumption 3** *For any given $\delta \in (0, 1)$, there exists $\theta_0 \in \mathbb{R}^p$ with $\|\theta_0\| = \Theta(1)$, that with probability at least $1 - \delta$*

$$\|\hat{\theta} - \theta_0\| < \phi(\delta, m),$$

*and $\phi(\delta, m)$ goes to 0 as $m$ goes to infinity. Also, there exist a threshold $M \in \mathbb{N}^+$ such that if $m > M$, $\phi(\delta, m)$ is a decreasing function of $\delta$ and $n$. Moreover,*

$$\min\left\{\frac{\theta_0^\top \theta^*}{\|\theta_0\|^2}, \theta_0^\top \theta^*, \|\theta_0\|\right\} > 0.$$

We will prove the following lemma as a cornerstone for our future proofs.

**Lemma 1** *Under Assumption 3, for any $\delta \in (0, 1)$, there exists a threshold $M \in \mathbb{N}^+$, and constants $0 < I_1 < I_2$, $0 < I_3 < I_4 < \alpha I_3$, $0 < I_5 < I_6$, such that if $m > M$, with probability at least $1 - \delta$ over the randomness of $S^{tr}$,*

$$\frac{\hat{\theta}^\top \theta^*}{\|\hat{\theta}\|^2} \in [I_1, I_2], \quad \hat{\theta}^\top \theta^* \in [I_3, I_4], \quad \|\hat{\theta}\| \in [I_5, I_6].$$

**Proof 1** *Under Assumption 3, we know $m \to \infty$ leads to $\hat{\theta} \to \theta_0$. In addition, for any $\delta \in (0, 1)$ there exists a threshold $M \in \mathbb{N}^+$ such that if $m > M$, $\phi(\delta, m)$ is a decreasing function of $\delta$ and $m$, which leads to*

$$\frac{\hat{\theta}^\top \theta^*}{\|\hat{\theta}\|^2} \in [\frac{\theta_0^\top \theta^*}{\|\theta_0\|^2} - \varepsilon, \frac{\theta_0^\top \theta^*}{\|\theta_0\|^2} + \varepsilon], \quad \hat{\theta}^\top \theta^* \in [\theta_0^\top \theta^* - \varepsilon, \theta_0^\top \theta^* + \varepsilon], \quad \|\hat{\theta}\| \in [\|\theta_0\| - \varepsilon, \|\theta_0\| + \varepsilon]$$

*for some small $\varepsilon > 0$ that makes the left end of the above intervals larger than 0 and $\theta_0^\top \theta^* + \varepsilon < \alpha(\theta_0^\top \theta^* - \varepsilon)$ hold for all $r_1, r_2, \sigma, m$ as long as $m > M$. Then, we set $I_i$'s accordingly to each value above.*

### C.2.1   Example Training $\hat{\theta}$

In this section, we provide one example to justify Assumption 3, i.e., $\hat{\theta} = \sum_{i=1}^m x_i^{tr} y_i^{tr}/m$, where the training set is drawn from an unperturbed Gaussian mixture, i.e., $x^{tr}|y^{tr} \sim \mathcal{N}(y^{tr} \cdot \theta^*, \sigma^2 I)$ and $y^{tr}$ follows a Bernoulli distribution $\mathbb{P}(y^{tr} = 1) = 1/2$. Directly following the analysis of Zhang et al. (2022), we have

$$\hat{\theta}^\top \theta^* = O_{\mathbb{P}}(\frac{1}{\sqrt{m}})\|\theta^*\| + \|\theta^*\|^2.$$

For $\|\hat{\theta}\|^2$, notice that

$$\hat{\theta} = \theta^* + \epsilon_m$$

where $\epsilon_m \sim \mathcal{N}(0, \frac{\sigma^2 I}{m})$. Then, we have

$$\|\hat{\theta}\|^2 = \|\theta^*\|^2 + 2\epsilon_m^\top \theta^* + \|\epsilon_m\|^2 = \|\theta^*\|^2 + \frac{p}{m} + O_{\mathbb{P}}(\frac{\sqrt{p}}{m}) + O_{\mathbb{P}}(\frac{1}{\sqrt{m}})\|\theta^*\|.$$

Given $p/m = O(1)$, combined with the form of classic concentration inequalities, one can verify this example satisfies Assumption 3.

### C.3 Background: ECE Calculation

Recall we denote $\hat{f}(x) = \max\{\hat{p}_{-1}(x), \hat{p}_1(x)\}$ and denote the predicition result $\hat{y} = \hat{C}(x)$. The definition of ECE is:
$$\text{ECE} = \mathbb{E}_{\hat{f}(x)}|\mathbb{P}[y = \hat{y} \mid \hat{f}(x) = p] - p|.$$

Notice that there are two cases.

- Case 1: $\hat{f}(x) = \hat{p}_1(x)$, by reparameterization, we have

$$\left|\mathbb{P}[y = \hat{y} \mid \hat{f}(x) = p] - p\right| = \left|\mathbb{P}[y = 1 \mid \hat{f}(x) = \frac{e^{2v}}{1 + e^{2v}}] - \frac{e^{2v}}{1 + e^{2v}}\right|$$
$$= \left|\mathbb{P}[y = 1 \mid \hat{\theta}^\top x = v] - \frac{e^{2v}}{1 + e^{2v}}\right|.$$

- Case 2: $\hat{f}(x) = \hat{p}_{-1}(x)$, by reparameterization, we have

$$\left|\mathbb{P}[y = \hat{y} \mid \hat{f}(x) = p] - p\right| = \left|\mathbb{P}[y = -1 \mid \hat{f}(x) = \frac{1}{1 + e^{2v}}] - \frac{1}{1 + e^{2v}}\right|$$
$$= \left|\mathbb{P}[y = -1 \mid \hat{\theta}^\top x = v] - \frac{1}{1 + e^{2v}}\right|$$
$$= \left|1 - \mathbb{P}[y = -1 \mid \hat{\theta}^\top x = v] - (1 - \frac{e^{2v}}{1 + e^{2v}})\right|$$
$$= \left|\mathbb{P}[y = 1 \mid \hat{\theta}^\top x = v] - \frac{e^{2v}}{1 + e^{2v}}\right|.$$

To summarize,

$$\text{ECE} = \mathbb{E}_{\hat{f}(x)}|\mathbb{P}[y = \hat{y} \mid \hat{f}(x) = p] - p| = \mathbb{E}_{\hat{\theta}^\top x}\left|\mathbb{P}[y = 1 \mid \hat{\theta}^\top x = v] - \frac{1}{1 + e^{-2v}}\right|.$$

**Temperature scaling.**
$$p_{-1}^T(x) = \frac{1}{e^{2 \cdot \hat{\theta}^\top x/T} + 1}, \quad p_1^T(x) = \frac{e^{2 \cdot \hat{\theta}^\top x/T}}{e^{2 \cdot \hat{\theta}^\top x/T} + 1}. \tag{15}$$

Thus,
$$\text{R-ECE} = \mathbb{E}_{\hat{\theta}^\top x}\left|\mathbb{P}[y = 1 \mid \hat{\theta}^\top x = vT] - \frac{1}{1 + e^{-2v}}\right|.$$

**Platt scaling.**
$$p_{-1}^{w,b}(x) = \frac{1}{e^{2w \cdot \hat{\theta}^\top x + 2b} + 1}, \quad p_1^{w,b}(x) = \frac{1}{e^{-2w \cdot \hat{\theta}^\top x - 2b} + 1}. \tag{16}$$

$$\text{ECE}_{w,b} = \mathbb{E}_{\hat{\theta}^\top x}\left|\mathbb{P}[y = 1 \mid w \cdot \hat{\theta}^\top x + b = v] - \frac{1}{1 + e^{-2v}}\right|.$$

**ECE calculation.** The distribution of $\hat{\theta}^\top x$ has the following properties.

- $\hat{\theta}^\top x | y = 1 \sim z\rho_1 \mathcal{N}(\hat{\theta}^\top \theta^*, \sigma^2 \|\hat{\theta}\|^2) \mathbf{1}\{\hat{\theta}^\top x \in \mathbb{B}(\hat{\theta}^\top \theta^*, r_1 \|\hat{\theta}\|) \cup \mathbb{B}(-\hat{\theta}^\top \theta^*, r_1 \|\hat{\theta}\|)\}$
  $+ (1 - z)\rho_2 \mathcal{N}(-\alpha \cdot \hat{\theta}^\top \theta^*, \sigma^2 \|\hat{\theta}\|^2)) \mathbf{1}\{x \in \mathbb{B}(\alpha \hat{\theta}^\top \theta^*, r_2 \|\hat{\theta}\|) \cup \mathbb{B}(-(\alpha \hat{\theta}^\top \theta^*, r_2 \|\hat{\theta}\|))\};$

- $\hat{\theta}^\top x | y = -1 \sim z\rho_1 \mathcal{N}(-\hat{\theta}^\top \theta^*, \sigma^2 \|\hat{\theta}\|^2) \mathbf{1}\{\hat{\theta}^\top x \in \mathbb{B}(\hat{\theta}^\top \theta^*, r_1 \|\hat{\theta}\|) \cup \mathbb{B}(-\hat{\theta}^\top \theta^*, r_1 \|\hat{\theta}\|)\}$
  $+ (1 - z)\rho_2 \mathcal{N}(\alpha \cdot \hat{\theta}^\top \theta^*, \sigma^2 \|\hat{\theta}\|^2)) \mathbf{1}\{\hat{\theta}^\top x \in \mathbb{B}(\alpha \hat{\theta}^\top \theta^*, r_2 \|\hat{\theta}\|) \cup \mathbb{B}(-\alpha \hat{\theta}^\top \theta^*, r_2 \|\hat{\theta}\|)\}.$

Now, we are ready to calculate ECE. Specifically, given $\hat{\theta}$, For notation simplicity, we denote $\mathcal{A} = \mathbb{B}(\hat{\theta}^\top \theta^*, r_1 \|\hat{\theta}\|) \cup \mathbb{B}(-\hat{\theta}^\top \theta^*, r_1 \|\hat{\theta}\|)$ and $\mathcal{B} = \mathbb{B}(\alpha \cdot \hat{\theta}^\top \theta^*, r_2 \|\hat{\theta}\|) \cup \mathbb{B}(-(\alpha \cdot \hat{\theta}^\top \theta^*, r_2 \|\hat{\theta}\|))$. Meanwhile, for simplicity, we choose $r_1, r_2$ such that $\rho_1 = \rho_2 = \rho$. This is always manageable and there exists infinitely many choices, we only require $S_1 \cap S_2 = \emptyset$ for any $S_1 \neq S_2$, $S_1, S_2 \in \{\mathbb{B}(\hat{\theta}^\top \theta^*, r_1 \|\hat{\theta}\|), \mathbb{B}(-\hat{\theta}^\top \theta^*, r_1 \|\hat{\theta}\|), \mathbb{B}(\alpha \hat{\theta}^\top \theta^*, r_2 \|\hat{\theta}\|), \mathbb{B}(-\alpha \hat{\theta}^\top \theta^*, r_2 \|\hat{\theta}\|)\}$. In able to achieve $\rho_1 = \rho_2 = \rho$, it only depends on $r_1/r_2$. Apparently, there exists a threshold $\phi > 0$ such that if $r_1$ $r_2$ are both smaller than $\phi$ (one can choose $r_1, r_2$ as functions of $\sigma$ with appropriate choosen $\sigma$), then $\mathcal{A} \cap \mathcal{B} = \emptyset$ can be achieved.

$$
\mathbb{P}[y = 1 \mid \hat{\theta}^\top x = v] = \frac{\mathbb{P}(\hat{\theta}^\top x = v \mid y = 1)}{\mathbb{P}(\hat{\theta}^\top x = v \mid y = 1) + \mathbb{P}(\hat{\theta}^\top x = v \mid y = -1)}
$$
$$
= \frac{1}{1 + \exp\left(\frac{-2\hat{\theta}^\top \theta^*}{\sigma^2 \|\hat{\theta}\|^2} \cdot v\right)} \mathbf{1}\{v \in \mathcal{A}\} + \frac{1}{1 + \exp\left(\frac{2\alpha\hat{\theta}^\top \theta^*}{\sigma^2 \|\hat{\theta}\|^2} \cdot v\right)} \mathbf{1}\{v \in \mathcal{B}\}
$$

### C.4 Proof of Theorem 1

#### C.4.1 Temperature Scaling Only

A simple reparameterization leads to:

$$
\text{R-ECE} = \mathbb{E}_{v = \hat{\theta}^\top x} \left| \frac{\mathbf{1}\{v \in \mathcal{A}\}}{1 + \exp\left(\frac{-2\hat{\theta}^\top \theta^*}{\sigma^2 \|\hat{\theta}\|^2} \cdot v\right)} + \frac{\mathbf{1}\{v \in \mathcal{B}\}}{1 + \exp\left(\frac{2\alpha\hat{\theta}^\top \theta^*}{\sigma^2 \|\hat{\theta}\|^2} \cdot v\right)} - \frac{1}{e^{-2v/T} + 1} \right|
$$

The lower bound contains two parts. We choose the threshold $\phi$ mentioned previously small enough such that $I_3 > \max\{r_1, r_2\}$. This can be achieved because $I_3$ is independent of $r_1, r_2$.

**Part I.** When $v \in \mathbb{B}(\hat{\theta}^\top \theta^*, r_1 \|\hat{\theta}\|) \subset \mathcal{A}$, and we know that $\mathcal{A} \cap \mathcal{B} = \emptyset$. Let us choose a threshold $\min\{I_1, I_3\}/\sigma^2 > \tau > 0$. Then for **any** $T > 0$, it must fall into one of the following three cases.

Case 1: $T^{-1}$ and $\hat{\theta}^\top \theta^*/(\sigma^2\|\hat{\theta}\|^2)$ are far: $T^{-1} - \hat{\theta}^\top\theta^*/(\sigma^2\|\hat{\theta}\|^2) > \tau$, recall $v = \hat{\theta}^\top x$, then

$$\mathbb{E}_{x\in\mathbb{B}(\hat{\theta}^\top\theta^*, r_1\|\hat{\theta}\|)} \left| \frac{1}{1+\exp\left(\frac{-2\hat{\theta}^\top\theta^*}{\sigma^2\|\hat{\theta}\|^2}\cdot v\right)} - \frac{1}{1+e^{-2v/T}} \right|$$

$$\geq \left[ \frac{1}{1+\exp\left(-2T^{-1}(\hat{\theta}^\top\theta^* - r_1)\right)} - \frac{1}{1+\exp\left(\frac{-2\hat{\theta}^\top\theta^*}{\sigma^2\|\hat{\theta}\|^2}(\hat{\theta}^\top\theta^* + r_1)\right)} \right] \mathbb{P}(x\in\mathbb{B}(\theta^*, r_1)))$$

$$\geq \frac{\beta}{2}\left[ \frac{1}{1+\exp\left(-2(\hat{\theta}^\top\theta^*/(\sigma^2\|\hat{\theta}\|^2) + \tau)(\hat{\theta}^\top\theta^* - r_1)\right)} - \frac{1}{1+\exp\left(\frac{-2\hat{\theta}^\top\theta^*}{\sigma^2\|\hat{\theta}\|^2}(\hat{\theta}^\top\theta^* + r_1)\right)} \right]$$

$$\geq \frac{\beta}{2}\left[ \frac{1}{1+\exp\left(-2(\hat{\theta}^\top\theta^*/(\sigma^2\|\hat{\theta}\|^2) + \tau)(\hat{\theta}^\top\theta^* - r_1)\right)} - \frac{1}{1+\exp\left(\frac{-2\hat{\theta}^\top\theta^*}{\sigma^2\|\hat{\theta}\|^2}(\hat{\theta}^\top\theta^* + r_1)\right)} \right]$$

$$\geq \frac{\beta}{2}\min_{c\in[I_1,I_2], d\in[I_3,I_4]}\left[ \frac{1}{1+\exp\left(-2(c/\sigma^2 + \tau)(d - r_1)\right)} - \frac{1}{1+\exp\left(-2c/\sigma^2(d + r_1)\right)} \right] := \beta_1$$

Case 2: $T^{-1}$ and $\hat{\theta}^\top\theta^*/(\sigma^2\|\hat{\theta}\|^2)$ are far: $\hat{\theta}^\top\theta^*/(\sigma^2\|\hat{\theta}\|^2) - T^{-1} > \tau$,

$$\mathbb{E}_{x\in\mathbb{B}(\hat{\theta}^\top\theta^*, r_1\|\hat{\theta}\|)} \left| \frac{1}{1+\exp\left(\frac{-2\hat{\theta}^\top\theta^*}{\sigma^2\|\hat{\theta}\|^2}\cdot v\right)} - \frac{1}{1+e^{-2v/T}} \right|$$

$$\geq \left[ \frac{1}{1+\exp\left(\frac{-2\hat{\theta}^\top\theta^*}{\sigma^2\|\hat{\theta}\|^2}(\hat{\theta}^\top\theta^* - r_1)\right)} - \frac{1}{1+\exp\left(-2T^{-1}(\hat{\theta}^\top\theta^* + r_1)\right)} \right]\cdot\frac{\beta}{2}$$

$$\geq \left[ \frac{1}{1+\exp\left(\frac{-2\hat{\theta}^\top\theta^*}{\sigma^2\|\hat{\theta}\|^2}(\hat{\theta}^\top\theta^* - r_1)\right)} - \frac{1}{1+\exp\left(-2(\hat{\theta}^\top\theta^*/(\sigma^2\|\hat{\theta}\|^2) - \tau)(\hat{\theta}^\top\theta^* + r_1)\right)} \right]\cdot\frac{\beta}{2}$$

$$\geq \frac{\beta}{2}\min_{c\in[I_1,I_2], d\in[I_3,I_4]}\left[ \frac{1}{1+\exp\left(-2c/\sigma^2(d - r_1)\right)} - \frac{1}{1+\exp\left(-2(c/\sigma^2 - \tau)(d + r_1)\right)} \right] := \beta_2$$

Case 3: When $T^{-1}$ and $\hat{\theta}^\top\theta^*/(\sigma^2\|\hat{\theta}\|^2)$ are close: $|T^{-1} - \hat{\theta}^\top\theta^*/(\sigma^2\|\hat{\theta}\|^2)| \leq \tau$, then when $v \in \mathbb{B}(-\alpha\hat{\theta}^\top\theta^*, r_2\|\hat{\theta}\|) \subset \mathcal{B}$. For small enough $\tau$ satisfying $\tau \leq 0.2(1-\alpha)I_1/\sigma^2$

$$\mathbb{E}_{v=\hat{\theta}^\top x\in\mathbb{B}(-\alpha\hat{\theta}^\top\theta^*, r_2\|\hat{\theta}\|)} \left| \frac{1}{1+\exp\left(\frac{2\alpha\hat{\theta}^\top\theta^*}{\sigma^2\|\hat{\theta}\|^2}\cdot v\right)} - \frac{1}{1+e^{-2v/T}} \right|$$

$$\geq \min_{a\in[\frac{-\alpha\hat{\theta}^\top\theta^*}{\sigma^2\|\hat{\theta}\|^2}, \frac{\hat{\theta}^\top\theta^*}{\sigma^2\|\hat{\theta}\|^2}+\tau]}\min_{v\in\mathbb{B}(-\alpha\hat{\theta}^\top\theta^*, r_2\|\hat{\theta}\|)} \frac{2v\exp(2va)}{(1+\exp(2av))^2}\left( \frac{2(1-\alpha)\hat{\theta}^\top\theta^*}{\sigma^2\|\hat{\theta}\|^2} - \tau \right)\frac{1-\beta}{2}$$

$$\geq \min_{a\in[-\frac{\alpha I_2}{\sigma^2}, \frac{I_2}{\sigma^2}+\tau]}\min_{v\in[\alpha I_3 - r_2 I_6, \alpha I_4 + r_2 I_6]} \frac{2v\exp(2va)}{(1+\exp(2av))^2}\left( 1.8(1-\alpha)\frac{I_1}{\sigma^2} \right)\frac{1-\beta}{2} := \beta_3$$

**Part III.** Combining together, we have

$$\text{R-ECE} \geq \min\{\beta_1, \beta_2, \beta_3\}.$$

Finally, we take $r_1 \leq \min\{0.1, \tau\sigma^2/I_1\}I_3$, which ensures $\beta_i > 0$ for all $i = 1, 2, 3$.

### C.4.2 Selective Calibration Only

We hope $\mathbb{E}[g(x) = 1] \geq \beta$. Let us first define $\mathcal{G} = \{\hat{\theta}^\top x : g(x) = 1\}$. Then, for **any** $g$, we have

$$
\mathbb{P}[y = 1 \mid \hat{\theta}^\top x = v, v \in \mathcal{G}]
$$

$$
= \frac{\mathbb{P}(\hat{\theta}^\top x = v, v \in \mathcal{G} \mid y = 1)}{\mathbb{P}(\hat{\theta}^\top x = v, v \in \mathcal{G} \mid y = 1) + \mathbb{P}(\hat{\theta}^\top x = v, v \in \mathcal{G} \mid y = -1)}
$$

$$
= \left( \frac{1}{1 + \exp\left( \frac{-2\hat{\theta}^\top \theta^*}{\sigma^2 \|\hat{\theta}\|^2} \cdot v \right)} \mathbf{1}\{v \in \mathcal{A}\} + \frac{1}{1 + \exp\left( \frac{2\alpha\hat{\theta}^\top \theta^*}{\sigma^2 \|\hat{\theta}\|^2} \cdot v \right)} \mathbf{1}\{v \in \mathcal{B}\} \right) \mathbf{1}\{v \in \mathcal{G}\}
$$

$$
= \frac{1}{1 + \exp\left( \frac{-2\hat{\theta}^\top \theta^*}{\sigma^2 \|\hat{\theta}\|^2} \cdot v \right)} \mathbf{1}\{v \in \mathcal{A} \cap \mathcal{G}\} + \frac{1}{1 + \exp\left( \frac{2\alpha\hat{\theta}^\top \theta^*}{\sigma^2 \|\hat{\theta}\|^2} \cdot v \right)} \mathbf{1}\{v \in \mathcal{B} \cap \mathcal{G}\}
$$

Then, by choosing small enough $\sigma$, such that $I_1/\sigma^2 > 1$, the corresponding ECE is:

$$
\text{ECE}_S = \mathbb{E}_{v = \hat{\theta}^\top x \mid \hat{\theta}^\top x \in \mathcal{G}} \left| \frac{1}{1 + \exp\left( \frac{-2\hat{\theta}^\top \theta^*}{\sigma^2 \|\hat{\theta}\|^2} \cdot v \right)} \mathbf{1}\{v \in \mathcal{A} \cap \mathcal{G}\} \right.
$$

$$
\left. + \frac{1}{1 + \exp\left( \frac{2\alpha\hat{\theta}^\top \theta^*}{\sigma^2 \|\hat{\theta}\|^2} \cdot v \right)} \mathbf{1}\{v \in \mathcal{B} \cap \mathcal{G}\} - \frac{1}{e^{-2v} + 1} \right|
$$

$$
\geq \mathbb{E}_{v = \hat{\theta}^\top x \mid \hat{\theta}^\top x \in \mathcal{G}} \left| \frac{1}{1 + \exp\left( \frac{-2\hat{\theta}^\top \theta^*}{\sigma^2 \|\hat{\theta}\|^2} \cdot v \right)} \mathbf{1}\{v \in \mathcal{A} \cap \mathcal{G}\} - \frac{1}{e^{-2v} + 1} \right|
$$

$$
+ \mathbb{E}_{v = \hat{\theta}^\top x \mid \hat{\theta}^\top x \in \mathcal{G}} \left| \frac{1}{1 + \exp\left( \frac{2\alpha\hat{\theta}^\top \theta^*}{\sigma^2 \|\hat{\theta}\|^2} \cdot v \right)} \mathbf{1}\{v \in \mathcal{B} \cap \mathcal{G}\} - \frac{1}{e^{-2v} + 1} \right|
$$

$$
\geq \lambda_1 \mathbb{P}(v \in \mathcal{A} \cap \mathcal{G} \mid v \in \mathcal{G}) + \lambda_2 \mathbb{P}(v \in \mathcal{B} \cap \mathcal{G} \mid v \in \mathcal{G}).
$$

Since $\mathbb{P}(v \in \mathcal{A} \cap \mathcal{G} \mid v \in \mathcal{G}) + \mathbb{P}(v \in \mathcal{B} \cap \mathcal{G} \mid v \in \mathcal{G}) = 1$, it is not hard to verify that

$$
\text{S-ECE} \geq \min\{\lambda_1, \lambda_2\}
$$

where

$$
\lambda_1 = \min_{a \in [1, \frac{I_2}{\sigma^2}]} \min_{v \in \mathcal{A}} \frac{2v \exp(2va)}{(1 + \exp(2av))^2} \left| \frac{I_1}{\sigma^2} - 1 \right|
$$

$$
\lambda_2 = \min_{a \in [-\frac{\alpha I_2}{\sigma^2}, 1]} \min_{v \in \mathcal{B}} \frac{2v \exp(2va)}{(1 + \exp(2av))^2} \left| \frac{\alpha I_1}{\sigma^2} - 1 \right|.
$$

### C.4.3 Selective Re-calibration

We choose $\mathcal{G} = \mathcal{B}$ and set $T^{-1} = \frac{\hat{\theta}^\top \theta^*}{\sigma^2 \|\hat{\theta}\|^2}$, then SR-ECE $= 0$. Thus, there exists appropriate choice of $g$ and $T$ such that

$$
\text{SR-ECE}(g, T) = 0.
$$

### C.5 Proof of Theorem 2

Usually, $\beta$ is much larger than $1 - \beta$, for example, $\beta = 90\%$. In this section, we impose the following assumption.

**Assumption 4** *The selector g will retain most of the probabilty mass in the sense that*

$$\beta > 2(1 - \beta).$$

Let us denote $\xi = \beta/2 - (1 - \beta)$ and $\xi$ is a positive constant. First, we have the following claim.

**Claim 5** *Under Assumption 4, if we further have*

$$\min_{v \in \mathbb{B}(\hat{\theta}^\top \theta^*, r_1 \|\hat{\theta}\|)} \left| \frac{1}{1 + \exp\left(\frac{-2\hat{\theta}^\top \theta^*}{\sigma^2 \|\hat{\theta}\|^2} \cdot v\right)} - \frac{1}{e^{-2v} + 1} \right|$$

$$> \max_{v \in \mathbb{B}(-\alpha\hat{\theta}^\top \theta^*, r_2 \|\hat{\theta}\|)} \left| \frac{1}{1 + \exp\left(\frac{2\alpha\hat{\theta}^\top \theta^*}{\sigma^2 \|\hat{\theta}\|^2} \cdot v\right)} - \frac{1}{e^{-2v} + 1} \right|,$$

*then for $g_1 = \arg\min_{g:\mathbb{E}[g(x)=1] \geq \beta}$ S-ECE, we have that $\mathbb{E}_{x \in \mathbb{B}(-\alpha\theta^*, r_2)}[g_1(x) = 1] = \mathbb{P}(x \in \mathbb{B}(-\alpha\theta^*, r_2))$.*

**Proof 2** *The proof is straightforward. We denote $O = \{x : x \in \mathbb{B}(-\alpha\theta^*, r_2), \ g(x) = 0\}$. We will prove that $\mathbb{P}(x \in O) = 0$.*

*If not, let us denote $\mathcal{P} = \mathbb{P}(x \in O) > 0$. Since we know $\beta > 2(1 - \beta)$, which means even if we "throw away" all the probability mass $1 - \beta$ by only setting points in $\mathbb{B}(\theta^*, r_1)$ with g value equals to 0, there will still be other remaining probability mass retained in $\mathbb{B}(\theta^*, r_1)$ with g value equals to 1. Then, there exists $g_2$ such that $g_2(x) = 1$ for all $x \in \mathbb{B}(-\alpha\theta^*, r_2)$ and leads to $\mathbb{P}(x \in \mathbb{B}(-\alpha\theta^*, r_2), g_2(x) = 1) = \mathbb{P}(x \in \mathbb{B}(-\alpha\theta^*, r_2), g_1(x) = 1) + \xi$ (enabled by the fact $\beta > 2(1 - \beta)$ ) and $\mathbb{P}(g_1(x) = 1) = \mathbb{P}(g_2(x) = 1)$ for $x \in \mathbb{B}(\theta^*, r_1) \cup \mathbb{B}(-\alpha\theta^*, r_2)$. Since*

$$\min_{v \in \mathbb{B}(\hat{\theta}^\top \theta^*, r_1 \|\hat{\theta}\|)} \left| \frac{1}{1 + \exp\left(\frac{-2\hat{\theta}^\top \theta^*}{\sigma^2 \|\hat{\theta}\|^2} \cdot v\right)} - \frac{1}{e^{-2v} + 1} \right|$$

$$> \max_{v \in \mathbb{B}(-\alpha\hat{\theta}^\top \theta^*, r_2 \|\hat{\theta}\|)} \left| \frac{1}{1 + \exp\left(\frac{2\alpha\hat{\theta}^\top \theta^*}{\sigma^2 \|\hat{\theta}\|^2} \cdot v\right)} - \frac{1}{e^{-2v} + 1} \right|,$$

*which means "throwing away" points in $\mathbb{B}(\alpha\theta^*, r_2)$ can more effectively lower the calibration error and we must have*

$$S\text{-}ECE(g_2) < S\text{-}ECE(g_1).$$

Next, we state how to set the parameters such that the condition in Claim 5 holds. As long as we choose $\sigma, r_1, r_2$ small enough, such that

$$\frac{1}{1 + \exp(-2I_1/\sigma^2(I_4 + r_1 I_6))} - \frac{1}{1 + \exp(-2I_1(I_3 - r_1 I_6))}$$

$$< \frac{1}{1 + \exp(-2(I_4 + r_2 I_6))} - \frac{1}{1 + \exp(2\alpha I_2/\sigma^2(I_4 + r_2 I_6))}$$

then,

$$\min_{v \in \mathbb{B}(\hat{\theta}^\top \theta^*, r_1 \|\hat{\theta}\|)} \left| \frac{1}{1 + \exp\left(\frac{-2\hat{\theta}^\top \theta^*}{\sigma^2 \|\hat{\theta}\|^2} \cdot v\right)} - \frac{1}{e^{-2v} + 1} \right|$$

$$> \max_{v \in \mathbb{B}(-\alpha\hat{\theta}^\top \theta^*, r_2 \|\hat{\theta}\|)} \left| \frac{1}{1 + \exp\left(\frac{2\alpha\hat{\theta}^\top \theta^*}{\sigma^2 \|\hat{\theta}\|^2} \cdot v\right)} - \frac{1}{e^{-2v} + 1} \right|,$$

Then, following similar derivation in Section C.4.1, we can prove with suitably chosen parameters $r_1, r_2, \sigma$, $\text{ECE}^{S \to T} > 0$.

Lastly, let us further prove $\text{ECE}^{T \to S} > 0$. We choose $r_1$ and $r_2$ small enough such that $v > 0$ for all $v \in \mathbb{B}(\hat{\theta}^\top \theta^*, r_1 \|\hat{\theta}\|) \cup \mathbb{B}(\alpha \hat{\theta}^\top \theta^*, r_2 \|\hat{\theta}\|)$ and $v < 0$ for all $v \in \mathbb{B}(-\hat{\theta}^\top \theta^*, r_1 \|\hat{\theta}\|) \cup \mathbb{B}(-\alpha \hat{\theta}^\top \theta^*, r_2 \|\hat{\theta}\|)$.

For $1/T \in [\frac{-\alpha \hat{\theta}^T \theta^*}{\sigma^2 \|\hat{\theta}\|}, \frac{\hat{\theta}^T \theta^*}{\sigma^2 \|\hat{\theta}\|}]$, we can calculate the derivative for R-ECE as the following:

$$
\text{R-ECE} = \mathbb{E}_{v \in \mathbb{B}(\hat{\theta}^\top \theta^*, r_1 \|\hat{\theta}\|)} \left[ \frac{1}{1 + \exp\left(\frac{-2\hat{\theta}^\top \theta^*}{\sigma^2 \|\hat{\theta}\|^2} \cdot v\right)} - \frac{1}{e^{-2v/T} + 1} \right]
$$

$$
+ \mathbb{E}_{v \in \mathbb{B}(-\hat{\theta}^\top \theta^*, r_1 \|\hat{\theta}\|)} \left[ -\frac{1}{1 + \exp\left(\frac{-2\hat{\theta}^\top \theta^*}{\sigma^2 \|\hat{\theta}\|^2} \cdot v\right)} + \frac{1}{e^{-2v/T} + 1} \right]
$$

$$
+ \mathbb{E}_{v \in \mathbb{B}(\alpha \hat{\theta}^\top \theta^*, r_2 \|\hat{\theta}\|)} \left[ -\frac{1}{1 + \exp\left(\frac{2\alpha \hat{\theta}^\top \theta^*}{\sigma^2 \|\hat{\theta}\|^2} \cdot v\right)} + \frac{1}{e^{-2v/T} + 1} \right]
$$

$$
+ \mathbb{E}_{v \in \mathbb{B}(-\alpha \hat{\theta}^\top \theta^*, r_2 \|\hat{\theta}\|)} \left[ \frac{1}{1 + \exp\left(\frac{2\alpha \hat{\theta}^\top \theta^*}{\sigma^2 \|\hat{\theta}\|^2} \cdot v\right)} - \frac{1}{e^{-2v/T} + 1} \right].
$$

Next, we take a derivative over $x = 1/T$ for $x \in [\frac{-\alpha \hat{\theta}^T \theta^*}{\sigma^2 \|\hat{\theta}\|}, \frac{\hat{\theta}^T \theta^*}{\sigma^2 \|\hat{\theta}\|}]$, which leads to

$$
\frac{d\text{R-ECE}}{dx} = -2\mathbb{E}_{v \in \mathbb{B}(\hat{\theta}^\top \theta^*, r_1 \|\hat{\theta}\|)} \left[ \frac{2ve^{2vx}}{(e^{2vx} + 1)^2} \right] + 2\mathbb{E}_{v \in \mathbb{B}(\alpha \hat{\theta}^\top \theta^*, r_2 \|\hat{\theta}\|)} \left[ \frac{2ve^{2vx}}{(e^{2vx} + 1)^2} \right]
$$

Consider the two values

$$
\frac{2ve^{2vx}}{(e^{2vx} + 1)^2}, \quad \frac{2\alpha ve^{2\alpha vx}}{(e^{2\alpha vx} + 1)^2},
$$

the ratio

$$
\frac{2\alpha ve^{2\alpha vx}}{(e^{2\alpha vx} + 1)^2} / \left[ \frac{2ve^{2vx}}{(e^{2vx} + 1)^2} \right] \to_{v \to 0} 2\alpha.
$$

That means if we take suitably small $r_1, r_2$ and let $\sigma \in [c_1, c_2]$ with appropriately chosen $c_1, c_2$

$$
\frac{d\text{R-ECE}}{dx} \bigg|_{x = \frac{\hat{\theta}^T \theta^*}{\sigma^2 \|\hat{\theta}\|}} < 0.
$$

Thus, we know the best choice of $1/T$ should not be equal to $\frac{\hat{\theta}^T \theta^*}{\sigma^2 \|\hat{\theta}\|}$. Then, notice $\beta > 2(1 - \beta)$, which means the probability mass in $\mathbb{B}(\hat{\theta}^T \theta^*, r_1 \|\hat{\theta}\|)$ cannot be all be "thrown away"; following similar derivation in Section C.4.1, we can prove with suitably chosen parameters $r_1, r_2, \sigma$, $\text{ECE}^{T \to S} > 0$.

