# OpenReview forum: "Improving Predictor Reliability with Selective Recalibration"
_TMLR — Accepted by TMLR_

### Review · Reviewer_K1dr · 2024-02-28

**Summary Of Contributions:**

This paper uses several concepts which I will begin by briefly summarizing.

Recalibration refers to techniques which modify the output of a classifier to ensure it is properly calibrated. Selective classification aims to only classify points where the classifier is confident enough (while obeying some coverage constraints) so as to increase classification accuracy, which is achieved by training a selection model which, given an input, determines if it should be classified or not. The recently proposed selective calibration uses a selection model for calibration purposes, i.e. to ensure that the classifier is properly calibrated on the selected points.

This paper proposes selective recalibration, which combines recalibration with selective calibration. In other words, the authors simultaneously use a selection model along with a small head which changes the output of a classifier to improve its calibration on selected datapoints. The authors propose a sensible objective to jointly train the recalibration and the selection networks given a pretrained classifier. The authors also argue that this simultaneous optimization is desirable by constructing an example where perfect calibration is unachievable when using only recalibration, only selective calibration, or even when using selective recalibration where the two networks are trained sequentially; yet perfect calibration is achievable when doing simultaneous optimization in selective recalibration.

Overall this is a good submission which is clearly written, well motivated, and which has good empirical results.

**Audience:**

Yes

**Broader Impact Concerns:**

I have no broader impact concerns.

**Claims And Evidence:**

Yes

**Requested Changes:**

Please see weaknesses section of my review.

**Strengths And Weaknesses:**

**Strengths**
1. The paper is well written and easy to follow.
2. The problem addressed in the paper is relevant to the community.
3. The proposed approach is simple, effective, works well in practice, and is convincingly motivated.

**Weaknesses**

4. Although the paper is mostly clear, I did find section 4.1 to be a bit ambiguous. For starters, eq 6 is stated without defining that $(x, y)$ is the batch. Then, the losses $l(f, \hat{g}, h, x, y)$ are not written in such a way that makes it clear for the reader what is fixed an what is being optimized, e.g. I think $l(\hat{g}, h; f, x, y)$ would be better notation. More importantly, I do not believe that $L_{set}$ as defined in eq 7, along with the choice of $l$ in eq 9 recover the loss used by Fisch et al., as claimed in the current manuscript; both because the scaling changes to $\sum_{i,j}\hat{g}(x_i)\hat{g}(x_j)$ and because $q$ is not used in the numerator in eq 6 (compare to eq 13 in Fisch et al.). Also, should eq be multiplied by $1/n^2$ instead of $1/n$? This makes it ambiguous if, in the experiments, the authors compare against the method as described in Fisch et al., or the slightly different version they describe in sec 4.1.2.
5. I do not believe that eq 10 provides an unbiased estimate of $(\beta - \mathbb{E}[\hat{g}(x)])^2$. I am aware that at test time you set the threshold so as to ensure coverage, but I still believe this should be discussed.
6. While the analysis in sec 6 is very interesting and convincingly motivates the need for joint training, the used classifier remains very simple. It would be good to discuss what might change in the setting where a neural network classifier is used instead. To be clear I'm not asking for theoretical results in the setting, just a discussion.
7. This paper mostly focuses on calibration, which can be understood as attempting to have proper quantification of aleatoric uncertainty. I think it would be appropriate to discuss methods aiming to also quantify epistemic uncertainty in the related work section, e.g. Bayesian neural networks, deep ensembles, and conformal prediction.
8. The organization of the paper, where the theoretical analysis in section 6 comes after the experiments in section 5, is bizarre.
9. Finally, a list of typos:
- First paragraph: "1% chance" -> "a 1% chance"
- Last paragraph of sec 2: cite Kumar et al. 2018 using \citet
- Equations are sometimes missing punctuation, e.g. missing period at the end of eq 2, and missing comma at the end of eq 4
- The sentence "Intuitively, to optimize..." after eq 4 is rather unclear, please rephrase
- eq 12: missing s subindex for f
- Missing indicator in $\hat{g}(x) \geq \tau$ in sec 4.4.1.
- Top of page 7: "ECE-1 and ECE-2" -> ECE_1 and ECE_2 for notational consistency
- Definition 1, while understandable, should use more formal notation, $x|y$ as written, is $x|y,z$, and the dependence on $y$ should be made clear in the notation, rather than in the paragraph following the definition.
- Please double check you cite the correct version of papers. I noticed you cite Fisch et al. (2022) as an arxiv preprint, but I believe tha's a TMLR paper. I did not go over all the citations, but please check them.

---

> ### Author Response · Authors · 2024-04-25
> **Author Response**
>
> We thank the reviewer for taking the time to review our manuscript.  We are glad you appreciated our presentation and results, and grateful for the detailed feedback on how the paper might be improved.  In general we agree with the reviewer's comments, and have included the relevant edits in our updated manuscript; below please find notes on individual points of feedback.
>
> Point 4 - Thank you for the suggestion on presenting the batch and loss functions, we agree with your suggestions and will update accordingly.  Regarding equation 9, please note that this equation is meant to correspond to equation 15 in Fisch et al.  Still, our presentation is slightly off, and have edited our equation to exactly match the first term in Fisch equation 15.  Regarding the experimental comparison, we adopt the code released with the Fisch paper to calculate SMMCE, so we are confident in that aspect of the comparison.
>
> Point 5 - It is true that this is not an unbiased estimator.  We have noted the fact that it is instead an asymptotically consistent estimator.
>
> Point 6 - Thank you, we have added a brief discussion along these lines to the paper. In order to show the effect of our method on the true population version of ECE and compare that with other methods (for example joint training v.s. sequential training), we need to have a tractable and closed form of expression of the population version of ECE. That is why we choose a simple classifier and a specific data generating distribution to do the analysis. If we choose a classifier as a neural network, the population version of ECE is no longer tractable mathematically.
>
> Point 7 - We have added references to more related work in uncertainty quantification including Bayesian neural networks, deep ensembles, and conformal prediction.
>
> Point  8 - We appreciate that analysis usually precedes experimental results.  Since we propose selective recalibration primarily as a practical solution, we believe empirical results are most important in our case.  Upon further consideration, we would like to maintain this structure.
>
> Point 9 - Thank you for the list of typos, these are all corrected in the updated manuscript.  The exception is updating the citations; we plan to do this before publishing a final manuscript, making sure each reference reflects the most updated archival venue for each paper.

---

> > ### Comment · Reviewer_K1dr · 2024-04-27
> > **Thanks for the rebuttal**
> >
> > Thank you for your replies and for the updates to the manuscript. I'm happy with everything, the one nitpick I still have is about point 5: while  your point about the estimator being asymptotically consistent is true, I don't think it's relevant, since the asymptotic behaviour here is with respect to the batch size which is likely to remain constant even if the dataset size is very large.

---

> > > ### Author Response · Authors · 2024-04-29
> > > **Author Response - Thank you**
> > >
> > > Thank you to the reviewer for considering our rebuttal and revised manuscript.  The point about the estimator is noted, we will reflect this in our next revision by removing the comment about consistency and being clear about the bias.

---

### Review · Reviewer_bH9Y · 2024-04-04

**Summary Of Contributions:**

This paper studies the problem of calibrating the predictions of a pre-trained model. To do so, the authors employ simultaneous selection and recalibration. While this combination on its own is not novel, the authors propose to jointly optimize the parameters of the selection and recalibration models rather than sequentially optimize them as has been studied in prior works. Methodologically, this modification is made quite simply by pushing the recalibrator model inside of the cross-entropy term.

The authors validate this intuitive modification with a selection of experiments on RxRx1 and CIFAR-100-C and show strong results indicating that their method is strictly better than prior approaches with respect to ECE. However, it is a little less clear in the current presentation of the paper exactly how the accuracy compares between the methods. This will be a necessary addition to fully recommend acceptance (more on this below).

The authors finish the paper by providing theoretical justification for their method in a toy setting. While I feel this gives a good intuition for why the approach works and may be useful to readers I do not see it as a strong contribution on its own, more of an aspect that strengthens the papers presentation.

**Audience:**

Yes

**Broader Impact Concerns:**

I do not have broader impact concerns about this work.

**Claims And Evidence:**

Yes

**Requested Changes:**

Addition of accuracy statistics in Table 1 and Table 2. Adjustments to preamble of equation (1) and re-write of equation (6).

**Strengths And Weaknesses:**

The paper and setting is well presented. I am not fully up to date on the latest research in the area, but from a cursory search and reading it seems the authors have covered the important related papers. The methodological improvement is straight-forward and easy to understand and the results that are generated are solid. On the basis of these strengths I am inclined to vote for accept on this work.

One critical missing piece of this work is the final accuracy of the models reported in Table 1 and Table 2. It does not seem necessary that the accuracy of the model would drop when the proposed method is applied, but having the concrete accuracy numbers would allow readers to better understand what trade-off exists, if any. I feel this is my strongest suggestion.

There are some minor presentation issues that ought to be fixed as well. Before equation (1) it is not made clear what $\hat{f}$ is precisely. In Equation (6) the authors are missing indices which make the equation improperly defined.

---

> ### Author Response · Authors · 2024-04-25
> **Author Response**
>
> Thank you for taking the time to review our manuscript and provide valuable feedback.
>
> With regards to the comment on including accuracy results, thank you for this suggestion.  We agree that while calibration is paramount in some settings, it is important to understand how our method influences the accuracy of the accepted samples.  We have added accuracy column to Table 1, and note that the values would be the same for Table 2 (which contains Brier Score results for the same experiment runs).
>
> We also appreciate the feedback on the presentation of equations 1 and 6.  Please note that $\hat{f}$ is defined in the first paragraph of Section 3 as the max class probability.  We have updated the paper to reflect the comment on equation 6.

---

> > ### Author Response · Authors · 2024-05-02
> > **Author Follow-up**
> >
> > We thank the reviewer again for taking the time to review our manuscript.  As the discussion period is coming to an end, please let us know if you have any further questions or concerns.  Thank you!

---

### Review · Reviewer_XaU7 · 2024-04-21

**Summary Of Contributions:**

This paper proposes a new framework, called “selective recalibration”, to improve the calibration of a pre-trained classifier. Post-hoc recalibration and selection approaches have been developed separately for this purpose, but the authors show that combining these two approaches could give a unique advantage and make further improvements. The authors demonstrated this advantage theoretically, under simple assumptions. Then, they also provide empirical evidence through extensive experiments on popular benchmarks.

**Audience:**

Yes

**Claims And Evidence:**

Yes

**Requested Changes:**

More empirical demonstrations for the current design choices and various feature extractors.

**Strengths And Weaknesses:**

### Pros

1. **Clarity**. Overall, the writing is clear and easy to follow. In addition, the organization of the main draft is well-established.
2. **Well-motivated problem and principled method**. Improving the reliability of pre-trained classifiers is an important problem for many practical scenarios. In addition, while the proposed method is simple and can be viewed as a naive combination of two existing approaches, the motivation is clear and demonstrated with the theoretical justification.

### Cons

1. **Limited empirical improvement**. The improvement from the proposed method is not significant compared to a simple baseline (”Selection with Confidence”), even though it requires an additional training process. For example, this baseline outperforms all the variants of the proposed method on the ImageNet dataset and shows comparable performances on the Camelyon 17 dataset. While the proposed method outperforms this baseline under distribution shift (Section 5.2), this baseline still significantly improves the calibration of the pre-trained model (e.g., ECE-1 on RxRx1: 0.308 (None) > Confidence (0.071) > Proposed (0.036)). The advantage of this simple baseline is more noticeable as the effectiveness of the proposed framework varies depending on the configurations and tested datasets.
2. **Limited justification on current design choices**. Currently, there are some fixed design choices such as the choice of feature extractor and the architecture of selector g. However, such a choice is not demonstrated yet; for example, it is unclear whether the proposed framework can be generalizable across different feature extractors on the same dataset.

### Editorial comments

1. Typo
    - ECE_q in Eq.1, while ECE-q in Section 5 and Tables.
2. To help the understanding of the reader, it would be better to provide more details about each dataset in the main text, including the statistics, used backbone, and the figures of examples.

---

> ### Author Response · Authors · 2024-04-25
> **Author Response**
>
> Thank you for your careful review of our paper.  We are glad that you appreciated the motivation of our work, as well as the clear presentation and extensive empirical results.  Below please find our replies to the two points that the reviewer raised in the Cons section.  Also, we appreciate the editorial comments and have reflected this feedback in our updated draft.
>
> Point 1 (confidence baseline)- As we mention in the beginning of section 5, all selection baselines are applied to the recalibrated model in order to make the strongest comparison.  Accordingly, in the experiment you are referencing, the confidence baseline is applied to a model that already has 0.055 ECE, meaning this baseline leads to a ~30\% increase in calibration error on the model to which it is applied, while our method leads to a ~50\% reduction on the same model.  This baseline is a heuristic strategy and will fail in cases where a model’s confident predictions are in fact poorly calibrated, such as in the RxRx1 and CIFAR-100 experiments, whereas under our approach this can be learned by $g$.
>
> Point 2 (design choices)- Thank you for your suggestion.  Since we are introducing the objective of selective recalibration, we are more focused on the high level design decisions related to how to compose selection and recalibration functions.  We believe that the most important design decisions in our work, besides choice of selection and recalibration method, are the choice of loss function and optimization procedure (joint vs. sequential).  We thus train all combinations of loss function and optimization procedure in all experiments, and these results are featured in the main paper.  As the reviewer noted, our experiments exploring these important design decisions are extensive, performed across 4 diverse feature extractors and datasets and more than 10 comparison methods.  Beyond these decisions, our aim was to make simple design choices, and show that these simple design choices can generalize to different settings.  Our experiments include medical and general image classification, supervised and self-supervised feature extractors, and ID and OOD test data, and there are no hyperparameters that are specific to a single experiment.
>
> However, we do understand that different design choices could have been made in other places.  For example, the architecture of $g$ can vary as long as it’s differentiable, and its input could even come directly from the image space.  There are also different possible coverage loss terms that could be used, and different feature extractors could be used for a given dataset.  We believe that exploring such design choices is outside the scope of this work, as our goal is to provide extensive initial experiments illustrating the effectiveness of our proposed approach and existing baselines by exploring our primary design choices of loss function and optimization method. Based on the reviewer's comment we have added further clarification of the design choices we choose to explore in our experiments.

---

> > ### Author Response · Authors · 2024-05-02
> > **Author Follow-up**
> >
> > We thank the reviewer again for the time and care taken in their review.  As the discussion period is coming to an end, please let us know if you have any further questions or concerns.  Thank you!

---

### Author Response · Authors · 2024-04-25
**Author Response - Thanks to Reviewers**

We thank the reviewers for the time and care taken to review our manuscript.  We are happy to know that all of the reviewers feel that we have supported our claims and clearly presented our evidence, and that TMLR's audience would be interested in the findings of our work.  We have submitted a revised manuscript with changes highlighted in blue.  We will also respond to each reviewer's questions individually, to note the specific changes we made based on each.

---

> ### Author Response · Authors · 2024-06-07
> **Follow Up**
>
> We thank the reviewers and AE again for their time and efforts, and look forward to hearing more.

---

### Decision · Action_Editor_Jzxq · 2024-07-14

**Recommendation:** Accept as is

**Comment:**

This paper addresses the problem of calibrating the predictions of a pre-trained classifier. The authors introduce a new technique, selective recalibration, which combines recalibration with selective calibration by jointly optimizing the parameters of both approaches. Several methods for this joint optimization are proposed, and they are easy to implement. The authors demonstrate its effectiveness through theoretical justification in a toy setting and empirical evidence on two benchmarks.

All three reviewers acknowledged the clarity of this paper, the significance of the problem, and the soundness of the proposed method. During the revision process, the authors successfully addressed the reviewers’ comments; consequently, all of the reviewers now agree to the acceptance. Therefore, AE recommends a clear acceptance of this paper.

**Audience:**

Yes

**Claims And Evidence:**

Yes